# Computational redesign of a hydrolase for nearly complete PET depolymerization at industrially relevant high-solids loading

Yinglu Cui [1,4] ✉, Yanchun Chen[1,4], Jinyuan Sun [1,2,4], Tong Zhu [1,4], Hua Pang[1], Chunli Li[1], Wen-Chao Geng [1,3] & Bian Wu [1] ✉

Biotechnological plastic recycling has emerged as a suitable option for addressing the pollution crisis. A major breakthrough in the biodegradation of poly(ethylene terephthalate) (PET) is achieved by using a LCC variant, which permits 90% conversion at an industrial level. Despite the achievements, its applications have been hampered by the remaining 10% of nonbiodegradable PET. Herein, we address current challenges by employing a computational strategy to engineer a hydrolase from the bacterium HR29. The redesigned variant, TurboPETase, outperforms other well-known PET hydrolases. Nearly complete depolymerization is accomplished in 8 h at a solids loading of 200 g kg⁻¹. Kinetic and structural analysis suggest that the improved performance may be attributed to a more flexible PET-binding groove that facilitates the targeting of more specific attack sites. Collectively, our results constitute a significant advance in understanding and engineering of industrially applicable polyester hydrolases, and provide guidance for further efforts on other polymer types.

Substantial efforts have been dedicated to discovering and engineering PET hydrolases over the past two decades, contributing to the development of PET biodegradation from the detection of trace amounts of released products to the attainment of high conversion[1–12]. When combating the challenges posed by enzymatic PET recycling, key parameters should be taken into consideration to attain an economically viable industrial process, in particular solids loading (> 150 g kg⁻¹) and product yield (> 90%)[13–15]. However, almost all studies obtain appreciable depolymerization yields at solids loadings lower than 30 g kg⁻¹, which is considerably below an industrially relevant level[15]. When increasing the solids loading to a high level, a reduction in the hydrolysis rate and conversion is commonly observed (referred to as "solids effect"), as demonstrated in other heterogeneous reactions such as the biomass conversion process[16].

A breakthrough was achieved with an engineered LCC variant (LCC^ICCG) that exhibited 90% depolymerization of pretreated PET waste

at an industrially relevant solids loading (200 g kg⁻¹)[17]. However, the 10% nonbiodegradable PET remained and reached a high crystallinity level of ~30% because of physical aging, hindering immediate reuse for PET depolymerization. A report issued by the National Association for PET Container Resources indicated that the postconsumer PET bottles collected for recycling reached 800 kilotons in the US[11], and this would generate over 80 kilotons of nonbiodegradable PET waste per year if the current biodegradation strategy was employed, posing a significant threat to the PET recycling economy. Increasing the extent of conversion from 90% to 99% would cut the minimum selling price (MSP) of TPA by 10% (US$0.2/kg) according to the TEA model proposed by the National Renewable Energy Laboratory in the US; thus, second to solids loading, the conversion level is the largest factor affecting MSP in the depolymerization process[14]. Both environmental and socioeconomic concerns have fueled intense interest in improving the degradation level to accelerate the transition towards the circular

[1]AIM Center, Institute of Microbiology, Chinese Academy of Sciences, Beijing, China. [2]University of Chinese Academy of Sciences, Beijing, China. [3]College of Chemistry, Nankai University, Tianjin, China. [4]These authors contributed equally: Yinglu Cui, Yanchun Chen, Jinyuan Sun, Tong Zhu. ✉ e-mail: cuiyinglu@im.ac.cn; wub@im.ac.cn

economy. According to the kinetic results of PET crystallization (Supplementary Fig. 1), physical aging can be vastly suppressed by decreasing the reaction temperature, but the catalytic efficiency of thermophilic PET hydrolases is concomitantly sacrificed[18,19], thus raising demands for biocatalysts with high depolymerization efficiency at low deformability temperatures of amorphous PET chains.

The last few years have witnessed impressive progress in tailoring natural enzymes by computational redesign strategies[20]. Inspired by the achievements in artificial intelligence for addressing the protein fitness landscape to probe hidden evolutionary information, we employed a computational strategy that incorporates a protein language model and force-field-based algorithms to engineer PET hydrolases with balanced thermostability and hydrolytic capacity. The redesigned variant (TurboPETase) derived from this campaign outperformed the most efficient PET hydrolases currently recognized in the field (LCC[21], LCC$^{ICCG}$[17], ICCG$^{I6M}$[22], BhrPETase[23], FastPETase[24], HotPETase[18], DepoPETase[25], $Ca$PETase$^{M9}$[26], and PES-H1$^{L92F/Q94Y}$[27]) over a range of temperatures (50 °C–65 °C). The extraordinary degradation performance afforded by TurboPETase allowed nearly complete depolymerization of post-consumer PET bottles in 8 h at a high industrially relevant substrate loading of 200 g kg$^{-1}$, with a maximum production rate of 61.3 g$_{hydrolyzed\ PET}$ L$^{-1}$ h$^{-1}$, addressing the challenge regarding residual nonbiodegradable PET waste.

## Results

### Computational redesign of an efficient PET hydrolase

PET hydrolases belong to serine-hydrolase family, a widely distributed group known for their relatively low substrate specificity. The degradation function of PET hydrolases is thought to preexist as a promiscuous function, which then evolves into a primary function[28,29]. Given our limited knowledge of how a sequence encodes catalytic functions in polymer-degrading enzymes[30], addressing the challenge by exploiting physics-based computational redesign and rational design approaches is a difficult task. Alternatively, a successful approach involves utilizing deep learning models to map the process from protein sequence to function, as demonstrated in many cases[31–33]. These models can capture hidden information indicating the improvement in polymer degradation along the evolution trajectory from the relative fitness of protein variants. To this end, we employed a language model trained on two datasets that involved approximately 26,000 homologous sequences of PET hydrolases across evolution, to predict the probability of amino acid variation from the evolutionary landscape (Fig. 1A). To select the template enzyme, we compared the activity of LCC, LCC$^{ICCG}$, BhrPETase, FastPETase, and HotPETase using amorphous PET films (Gf-PET, from the supplier Goodfellow) across a range of temperatures (Supplementary Fig. 2). BhrPETase shares a high sequence identity of 94% with LCC, whereas LCC$^{ICCG}$ represents a variant of LCC characterized by four amino acid substituents (F243I/D238C/S283C/Y127G). Both FastPETase and HotPETase are derivatives engineered from $Is$PETase (Supplementary Fig. 3). At 65 °C, BhrPETase and LCC$^{ICCG}$ exhibited the highest catalytic performance of all PET hydrolases and temperatures tested and were chosen as the inputs. A Transformer encoder was used to process input amino acid sequences with absolute position embedding. Residue positions were sorted by the mean of the logits of 19 mutations assigned to the wild type amino acid at each position. The top ten amino acid positions with the highest average scores of each model were selected. After duplicated amino acid positions were removed, 18 amino acid positions at which the wild-type residues fit less well than potential substitutions were obtained (Supplementary Table 1).

According to the crystal structures of BhrPETase (PDB code: 7EOA) and LCC$^{ICCG}$ S165A in complex with MHET (PDB code: 7VVE[34]), 7 of 18 generated amino acid positions (W104, H164, M166, W190, H191, H218, and F/I243) were suggested to be embedded in a PET-binding groove. Due to the relatively higher thermostability of

BhrPETase, we subjected the 7 positions on this enzyme to generate 34 variants. Chen et al. reported that the Ser214/Ile218 double mutants of several $Is$PETase-homologous enzymes showed enhanced PET hydrolysis activity (by at least 1.3-fold) but vastly decreased $T_m$ values (by approximately 10 °C)[35]. Since His218 (corresponding to Ser214 in $Is$PETase) was involved in our predicted candidates, the hydrolytic performance of the H218S/F222I variant was also explored (Supplementary Table 2). Among the variants, BhrPETase$^{H218S/F222I}$ (referred to as BhrPETase M2) resulted in the highest improvements with a 1.7-fold increase in PET-hydrolytic activity at 65 °C, rendering this variant the new template for the second-round accumulation of the mutations at the remaining 6 positions. During the 2nd accumulation stage, only mutations at W104 and F243 positions (W104L, W104S, W104H, W104G, F243I, F243T, and F243G) on M2 exhibited improved hydrolytic activities (by 10% to 34%), albeit with significant decrease in thermostability (Supplementary Table 3).

Empirical data have suggested that a $T_m$ 15 °C higher than the $T_{opt}$ is necessary for catalyst longevity[27]. Hence, a PET hydrolase with a $T_m$ at least over 80 °C is preferred for efficient PET degradation. Notably, the M2 variant exhibited a melting temperature of 85 °C, which was 11 °C lower than that of the wild-type enzyme. The active mutations at the W104 positions on M2 reduced the stability even further, with $T_m$ values ranging from 71.5 °C to 75.5 °C. The nonnegligible decrease in stability limited further combination, and compensatory mutations needed to be introduced first to suppress the deleterious effects. Therefore, we applied our previously devised GRAPE strategy[36], which employs four complementary algorithms, namely, FoldX[37] (force field energy function), Rosetta_cartesian_ddg[38] (force field energy function), ABACUS[39] (statistical energy function) and DDD[40] (force field energy function), to design stabilizing mutations to compensate and buffer the destabilizing mutations (Fig. 1B). Upon experimental validation, 3 beneficial variants (A209R, D238K, and A251C-A281C) resulted in improved thermostability without compromising the activity (Supplementary Table 4). We added the stabilizing variants to the M2 variant using a stepwise combining strategy and resulted in a BhrPETase M6 variant (BhrPETase$^{H218S/F222I/A209R/D238K/A251C/A281C}$), which exhibited a restored melting temperature of 97 °C without sacrificing activity. Subsequently, active mutations at the W104 and F243 positions were combinatorically assembled and accumulated onto the thermostable M6 variant, generating 12 new variants (Fig. 1C and Supplementary Table 5). After a consideration of both depolymerization performance and thermostability, the best combination variant, BhrPETase$^{H218S/F222I/A209R/D238K/A251C/A281C/W104L/F243T}$ (referred to as TurboPETase), was selected with a $T_m$ of 84 °C and a 3.4-fold improvement in PET-specific activity towards GF-PET films compared to wild-type BhrPETase (Fig. 1D). In industrial scenarios, enzymatic hydrolysis is typically conducted in water rather than in a dedicated buffered system to simplify downstream processing. Aligned with the industrial preferences, we also evaluated the 12 variants under a low buffer concentration. The results revealed that with the exception M6$^{W104G/F243T}$, all variants largely retained their degrading activity in 100 mM potassium phosphate buffer (Supplementary Fig. 4 and Supplementary Table 6). TurboPETase consistently outperformed the other variants under both low and high buffer concentrations, making it the most effective variant for further investigation.

The depolymerization performance of TurboPETase was subsequently evaluated across a temperature range of 50–65 °C with respect to other PET hydrolases (Fig. 2A, B, and Supplementary Fig. 5). At 65 °C, TurboPETase exhibited the highest overall degradation of all PET hydrolases and temperatures tested, releasing 29.66 mM of the products (sum of BHET, MHET, and TPA) in 3 h. Counterparts like BhrPETase, LCC, LCC$^{ICCG}$, ICCG$^{I6M}$, and PES-H1$^{L92F/Q94Y}$ which exhibit high degradation performance at elevated temperatures, rendered hydrolytic activity 1.8-, 4.7-, 2.1-, 2.0-, and 19-fold lower than that of TurboPETase, respectively. At suboptimal temperatures, TurboPETase

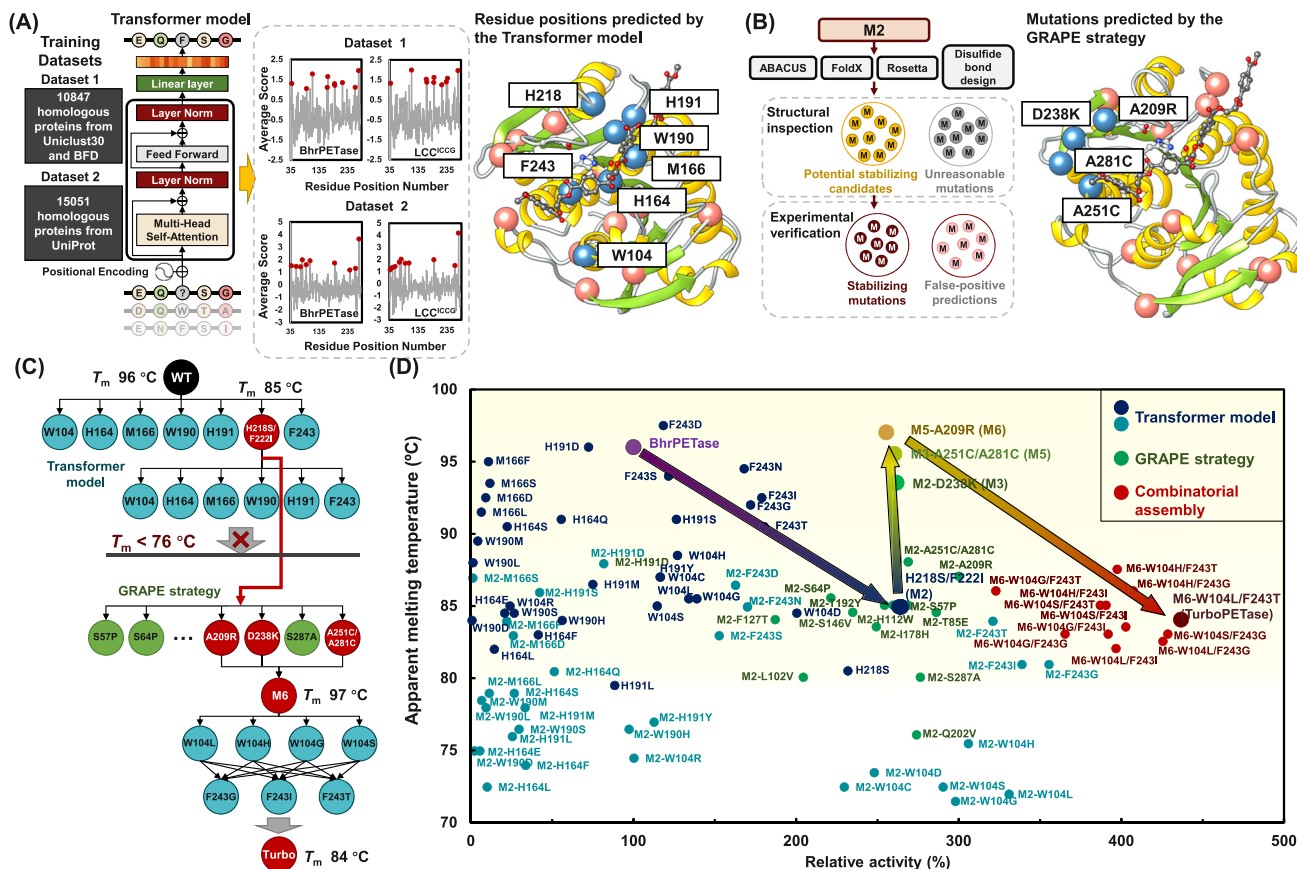

**Fig. 1 | Schematic representation of the redesign of PET hydrolases using the hybrid computational method. A** Potentially beneficial mutations are predicted with the Transformer model trained on two datasets (left panel). The top ten predicted amino acid positions (red circles) were selected. Removal of duplicate positions resulted in 18 residue positions, of which 7 positions were suggested to be located at the PET binding site (right panel). The Cα atoms of these amino acid positions are shown as blue spheres, whereas the 11 remaining positions are shown as coral spheres. PET and the catalytic triad are shown as ball-and-stick representations. **B** Schematic representation of the GRAPE strategy. Four algorithms were employed to predict the stabilizing mutations (left panel). After structural filtering, the potential stabilizing candidates were verified experimentally (right panel). The Cα atoms of the mutations with improved thermostability are shown as blue spheres, whereas other experimentally verified mutations are shown as coral spheres. **C** Schematic representation of the accumulation step. Mutations at the 7 residue positions located at the PET binding site were added to the wild type

enzyme (BhrPETase), resulting in the best hit BhrPETase[H218S/F222I] (M2). Adding the mutations at the remaining 6 positions to M2 led to active variants with dramatically decreased stability, and therefore, the GRAPE strategy was employed. The stabilizing mutations were combined into M2 and resulted in the thermostable variant M6, which can compensate for the destabilizing mutations in further accumulation steps. The active mutations at the W104 and F243 positions were combinatorially assembled on M6, leading to the final mutant TurboPETase. **D** Thermostability and relative PET-degrading activity of BhrPETase and the variants. The activities were homogenized according to BhrPETase. Data are from one independent experiment. Mutations predicted by the Transformer model and the GRAPE strategy are colored blue and green, respectively. The double mutants at the W104 and F243 positions on M6 are shown in red. Arrows represent the accumulation path. The yellow area represents the optimal melting temperature range for PET hydrolases to enable efficient depolymerization at 65 °C.

consistently outperformed other PET hydrolases, albeit the decreased hydrolytic activity. At the optimal temperature of HotPETase and *Ca*PETase[M9] of 60 °C, TurboPETase generated 11.73 mM monomer products in 3 h, whereas HotPETase and *Ca*PETase[M9] produced 4.32 mM and 0.60 mM monomers, which were 1.7-, and 18-fold lower than that of TurboPETase. When subjected to 50 °C, TurboPETase registered hydrolytic efficiencies exceeding those of FastPETase and DepoPETase by 43% and 59%, respectively. In light of the lower enzyme concentrations employed in certain reports, we recalibrated our reactions to mirror these conditions to ensure a fairer comparison (Fig. 2C−E). Under their reported reaction conditions, the reaction rates of LCC[ICCG], HotPETase, and FastPETase were 1.8-, 4.9-, and 1.0-fold lower, respectively, than that of TurboPETase. Recently, similar experiments were conducted to evaluate the degradation performance of FastPETase, HotPETase, PES-H1[L92F/Q94Y] and LCC[ICCG] under industrial conditions[41]. Their results demonstrated that HotPETase exhibited a higher specific activity compared to PES-H1[L92F/Q94Y] at a low substrate loading of 2 g L[−1]. However, as the substrate concentrations

increased to 16.5% (w/w) and 20% (w/w), the PET conversion of Hot-PETase were significantly lower than those of PES-H1[L92F/Q94Y]. This reduced efficiency at higher substrate loadings is suggested to be attributed to the limited thermostability and other catalytic properties of HotPETase, such as product inhibition. Despite the good performance of PES-H1[L92F/Q94Y] under industrially relevant conditions, the final depolymerization was still lower than that achieved by LCC[ICCG]. Given the reported enhancements of PES-H1[L92F/Q94Y] in 1 M potassium phosphate buffer, we also compared TurboPETase with PES-H1[L92F/Q94Y] under elevated buffer concentration, with TurboPETase still outpacing, yielding up to 2.5 times the degradation products of PES-H1[L92F/Q94Y] at 65 °C (Supplementary Fig. 6). To evaluate the long-time stability of TurboPETase, we extended the reaction time. The results demonstrated continuous enzyme performance of TurboPETase at elevated temperatures, as it produced 130 mM of soluble monomer products over 12 h at 65 °C, whereas the released products with BhrPETase, LCC, LCC[ICCG] were substantially lower (Supplementary Fig. 7). The degradation results highlight the substantially superior hydrolytic

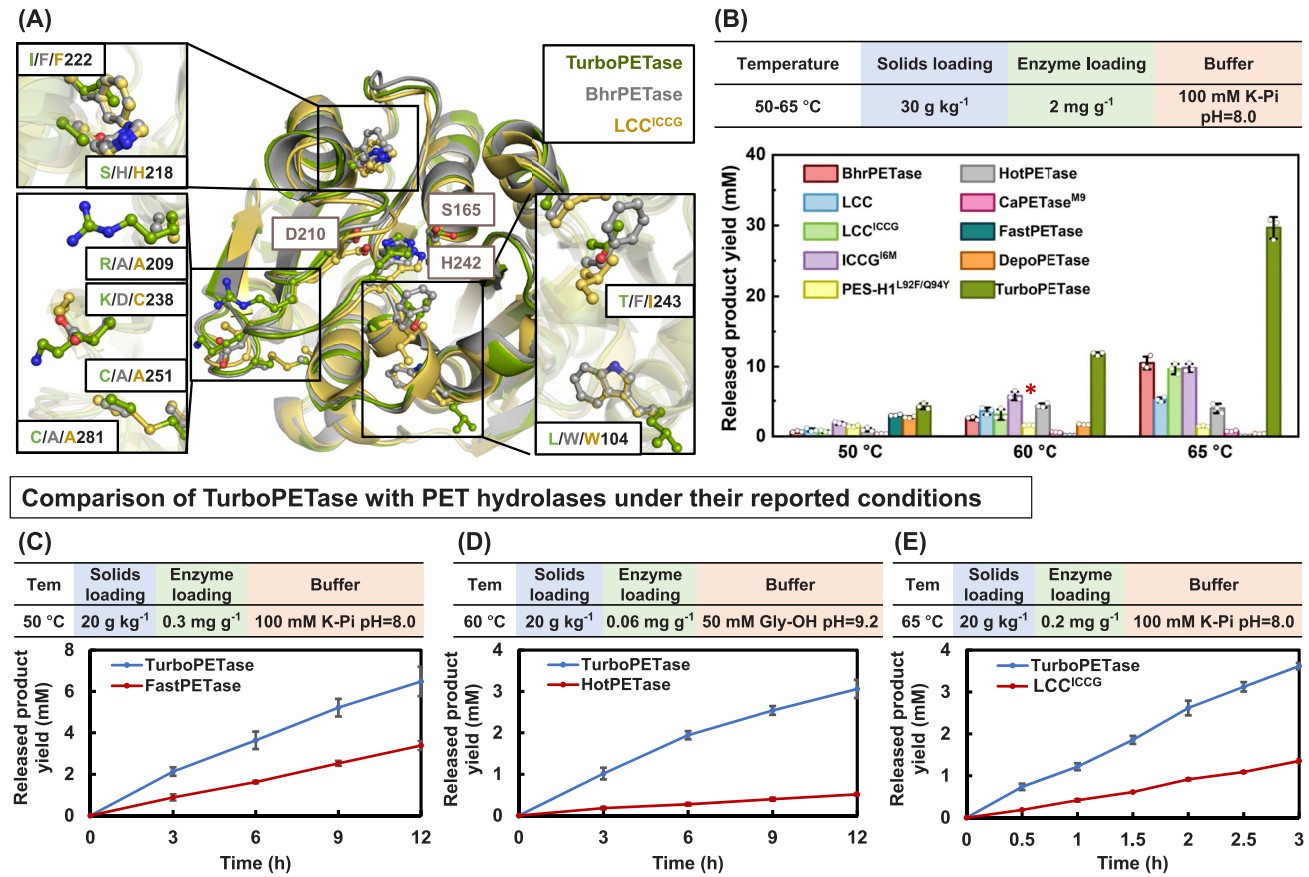

**Fig. 2 | Comparison of depolymerization performance of TurboPETase with other known PET hydrolases. A** Location of the beneficial mutations in Turbo-PETase and its counterparts. TurboPETase, BhrPETase and LCC$^{ICCG}$ are shown in green, grey and yellow, respectively. Residues are shown in ball and stick representations. **B** Comparison of the PET-hydrolytic activity of TurboPETase and other PET hydrolases towards Gf-PET films at temperatures ranging from 50 to 65 °C using 30 g kg$^{-1}$ solids loading and 2 mg$_{emzyme}$ g$_{PET}$$^{-1}$ enzyme loading in 100 mM potassium phosphate buffer, pH 8.0. The bar chart shows the mean depolymerization after 3 h of reaction. Data are presented as mean ± s.d. (*n* = 3 biologically independent experiments). The circles represent the individual numbers. *In the investigation conducted by Arnal et al.[41], the depolymerization efficiency of PES-H1$^{L92F/Q94Y}$ notably surpassed that of HotPETase, yet it remained inferior to LCC$^{ICCG}$, under industrially relevant substrate loadings of 16.5% (w/w) and 20% (w/w). **C** Time

course reactions of TurboPETase and FastPETase towards Gf-PET films at 50 °C under its reported reaction conditions (0.3 mg$_{emzyme}$ g$_{PET}$$^{-1}$ enzyme loading, 20 g kg$^{-1}$ solids loading, 100 mM potassium phosphate buffer, pH 8.0). Data are presented as mean ± s.d. (*n* = 3 biologically independent experiments). **D** Time course reactions of TurboPETase and HotPETase towards Gf-PET films at 60 °C under its reported reaction conditions (0.06 mg$_{emzyme}$ g$_{PET}$$^{-1}$ enzyme loading, 20 g kg$^{-1}$ solids loading, 50 mM Glycine-NaOH buffer, pH 9.2). Data are presented as mean ± s.d. (*n* = 3 biologically independent experiments). **E** Time course reactions of TurboPETase and LCC$^{ICCG}$ towards Gf-PET films at 65 °C using its reported enzyme loading of 0.2 mg$_{emzyme}$ g$_{PET}$$^{-1}$ in 100 mM potassium phosphate buffer (pH = 8.0). The solids loading was 20 g kg$^{-1}$. Data are presented as mean ± s.d. (*n* = 3 biologically independent experiments). Source data are provided as a Source Data file.

performance of TurboPETase across various temperatures and other reaction conditions tested.

## Michaelis–Menten approach and structural analysis

To interpret the enhanced hydrolysis performance, the kinetics were analysed through conventional Michaelis–Menten model and inverse Michaelis–Menten model. As shown in Fig. 3A, the curves exhibited near-linear relationships for the initial rate and substrate loading under all conditions for TurboPETase, BhrPETase and LCC$^{ICCG}$. Conventional saturation behaviour was not observed because even the lowest enzyme concentrations used here (0.12 µM) were too high for the conventional approach to be valid, which indicates the very fast rates of dissociation of the enzymes from the PET surface. Additionally, we conducted a kinetic comparison for soluble substrates (MHET and *p*NPB) as detailed in Supplementary Fig. 8 and Supplementary Table 7. The catalytic efficiency of TurboPETase towards MHET showed only a modest enhancement relative to PET (with a 32% increase), suggesting that there may be other factors, potentially increased adsorption to the surface, contributing to the amplified degradation efficiency of PET. Conversely, the slight decrement in TurboPETase's $k_{cat}$ for *p*NPB,

accompanied by a reduced binding affinity, inferred potential changes in the substrate binding domain, rendering it less conducive for other small molecule interactions.

Although the hydrolysis of PET could not meet the criteria for the conventional approach, the inverse Michaelis–Menten model was more applicable. The inverse Michaelis–Menten equation has been successfully employed to study the kinetics of heterogeneous enzymes such as cellulases, by which the catalytic efficacy against accessible attack sites on the polymer surface can be estimated[42–45]. As a typical surface erosion process, enzymatic hydrolysis of PET can hardly permeate the inner core of the polymer, resulting in a limited number of superficial ester bonds (also termed attack sites) being accessed even if the enzymes are in great excess. Saturation thus occurs when all sites on the surface become occupied, and the excess enzyme molecules accumulate in the solvent[42]. It should be noted that not all adsorption sites are competent for catalytic conversion, non-specific adsorption also accounts for a considerable proportion[44]. We measured free enzyme concentrations, $E_{free}$, and converted it to substrate coverage, $\Gamma = (E_{tot} - E_{free})/S_{PET}$ to calculated the total adsorption of the enzymes to PET surface (Supplementary Fig. 9 and

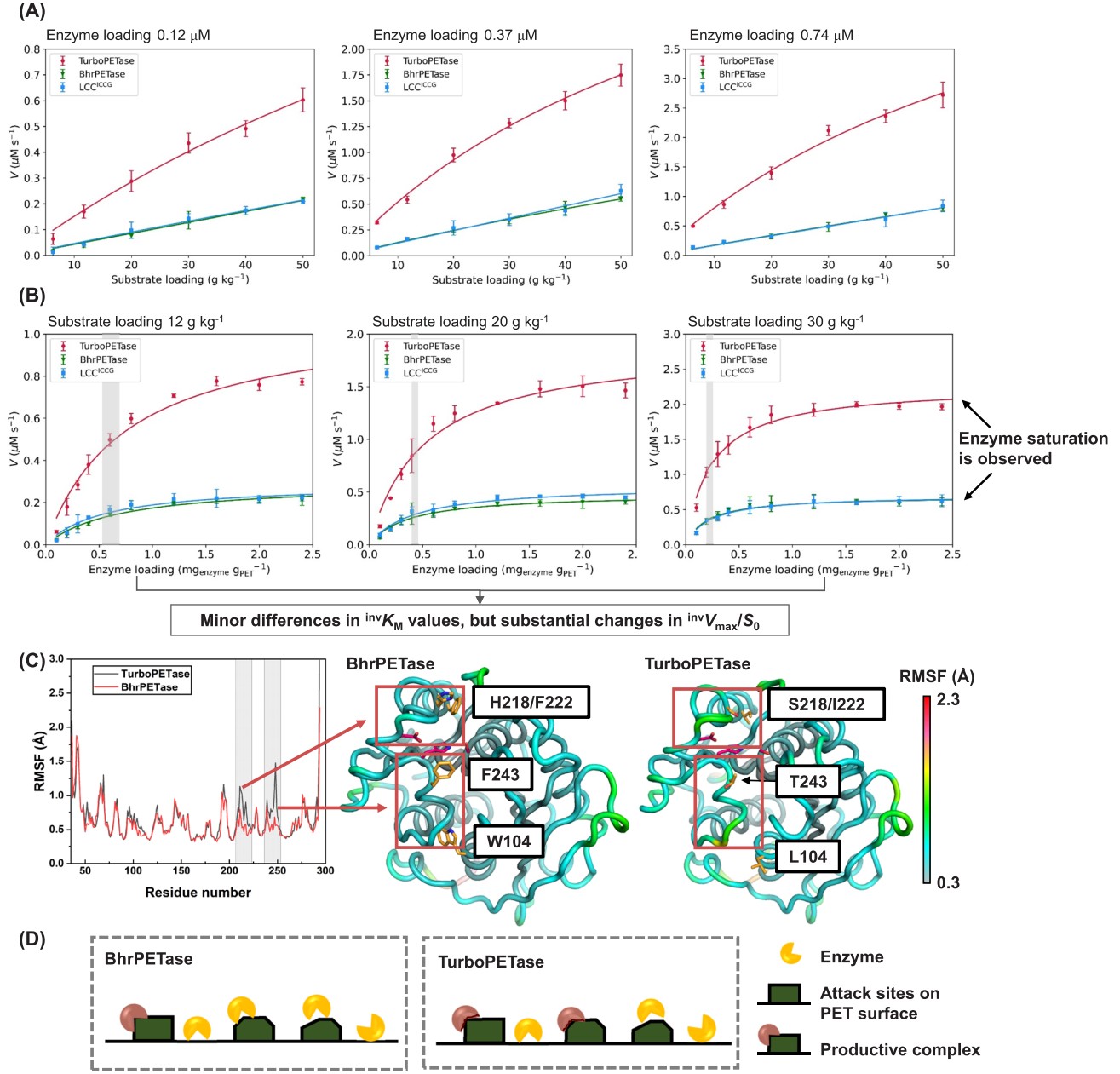

**Fig. 3 | Kinetic parameters and structural analysis of TurboPETase.**
**A** Conventional and **B** inverse Michaelis–Menten plots of TurboPETase, BhrPETase, and LCC$^{ICCG}$ at 65 °C in 100 mM potassium phosphate buffer, pH 8.0. Data are presented as mean ± s.d. ($n$ = 3 biologically independent experiments). Source data are provided as a Source Data file. **C** RMSF values (in Å) for all Cα atoms of BhrPETase and TurboPETase during the MD simulations, indicating global changes in protein flexibility, especially for the PET-binding groove. The H218S/F222I, W104L, and F243T mutations are shown as orange atoms, whereas the catalytic triad is shown as hot pink atoms. **D** Schematic diagram of PET depolymerization processes. Compared to the wild-type enzyme, TurboPETase may be more promiscuous due to the increased flexibility along the protein PET-binding groove, which allows it to attack different surface structures.

Supplementary Table 8). $\Gamma_{max}$ represents the (apparent) saturated bound enzymes, whereas $K_d$ is a dissociation constant[46]. At lower enzyme concentrations, TurboPETase exhibited a higher adsorption to the PET surface compared to its counterparts. However, upon reaching saturation, similar maximum adsorption capacities ($\Gamma_{max}$) were observed, underscoring a consistent overall binding potential across the enzymes. Comparisons of these enzymes could be expanded by considering specific changes in inverse Michaelis–Menten parameters, which can reflect the binding capability to the attack sites of the PET surface. As shown in Fig. 3B and Table 1, the $^{inv}K_M$ values revealed marginal differences between TurboPETase, BhrPETase, and LCC$^{ICCG}$, especially for the values at 30 g kg$^{-1}$ substrate loading, indicating that the specific adsorption capacity of TurboPETase at the attack sites on

the PET surface was not impaired. It's noteworthy that TurboPETase exhibited a 2.1-fold increase in $^{inv}V_{max}/^{mass}S_O$ compared with BhrPETase and LCC$^{ICCG}$. The inverse parameter, maximal reaction velocity per available reactive site ($^{inv}V_{max}/^{mass}S_O$), specifies the rate when all attack sites are covered with the enzyme[42]. Since no substantial differences in $^{inv}K_M$ and $\Gamma_{max}$ values were observed among the enzymes, the elevated $^{inv}V_{max}/^{mass}S_O$ values may imply a broadened targeting of TurboPETase towards specific attack sites when the enzymes maintained a stable overall adsorption level. Consequently, we presumed that the enhanced depolymerization performance of TurboPETase may rely, at least in part, on the enhanced ability to attack a broader spectrum of specific attack sites that can be hydrolysed to form a productive complex.

**Table 1 | Kinetic parameters of TurboPETase, BhrPETase, and LCC$^{ICCG}$ derived from the inverse Michaelis–Menten experiments**

| Parameters | $^{inv}K_M$ (mg$_{enzyme}$ g$_{PET}$$^{-1}$) | | | $^{inv}V_{max}$/S (µmol g$^{-1}$ s$^{-1}$) | | |
|---|---|---|---|---|---|---|
| | Substrate loading | | | | | |
| | 12 g kg$^{-1}$ | 20 g kg$^{-1}$ | 30 g kg$^{-1}$ | 12 g kg$^{-1}$ | 20 g kg$^{-1}$ | 30 g kg$^{-1}$ |
| TurboPETase | 0.67 (± 0.13) | 0.46 (± 0.08) | 0.24 (± 0.03) | 0.081 (± 0.006) | 0.082 (± 0.006) | 0.071 (± 0.002) |
| BhrPETase | 0.69 (± 0.16) | 0.40 (± 0.06) | 0.20 (± 0.04) | 0.020 (± 0.002) | 0.022 (± 0.001) | 0.022 (± 0.001) |
| LCC$^{ICCG}$ | 0.54 (± 0.12) | 0.42 (± 0.07) | 0.23 (± 0.03) | 0.021 (± 0.002) | 0.027 (± 0.001) | 0.023 (± 0.001) |

An in-depth structural analysis also helps us to understand the molecular underpinnings of the performance improvements. According to the model of TurboPETase predicted by AlphaFold2[47] and subsequent molecular dynamics (MD) simulations, the improved performance of TurboPETase may be attributed to the following key aspects: improved flexibility of the substrate binding cleft (H218S/F222I, W104L and F243T), optimized charge-charge interactions at the protein surface (A209R and D238K), and introduction of a disulfide bond (A251C-A281C). A209R, D238K, and the disulfide bond A251C-A281C are suggested to primarily contribute to improving thermostability while maintaining activity (Supplementary Figs. 10 and 11), whereas the enhanced hydrolysis efficacy may be attributed to substitutions in proximity to the active sites. H218 is suggested to form an intimate packing with the conserved W190 in analogous enzymes. Chen et al. found that PET hydrolytic activity could benefit from a more flexible active site in the H214S/F218I double mutant (corresponds to H218S/F222I in BhrPETase)[35]. In the present study, MD simulations of the apo form of TurboPETase revealed an expanded rotational freedom of W190 endowed by the H218S/F222I mutation (Supplementary Fig. 12), which is consistent with the observation of diverse conformations of the corresponding W156 of *Is*PETase[35]. When binding to the PET, the wobbling of W190 is curtailed and anchored by the π-π interactions with the PET substrate. This flexibility of the PET binding cleft was further enhanced by the synergistic interactions conferred by the addition of W104L and F243T, as revealed by the Cα root-mean-square fluctuation (RMSF) results (Fig. 3C). W104 is previously reported to pack against the adjacent P248 to stabilize the P248-situated β8-α6 loop[34]. In the redesigned TurboPETase, the relinquishment of this interaction by leucine substitution may engender increased conformational malleability within the loop region (N246-A250), as demonstrated by the largely reduced cross-correlation of these regions (Supplementary Fig. 13). For another substitution F243T in the PET binding cleft, the steric profile of F243 appears to dictate a more peripheral binding locus for PET. Yet, its mutation to threonine, armed with a less pronounced steric feature, may release the space for PET binding with a more-flexible state. More importantly, without the steric profile of the aromatic ring, T243 may beckon PET deeper into the cleft, drawing the substrate's labile carbonyl closer to the catalytic serine, with the interstitial distances contracting from 4.88 ± 0.51 Å to 4.15 ± 0.37 Å (Supplementary Fig. 14). Concurrently, the enhanced flexibility might compromise the protein's stability, which is consistent with the observed decrease in the melting temperatures of the single point mutations. Synthesizing our structural analysis with the kinetic data, we postulated that the greatly increased flexibility along the PET-binding groove may provide more space to accommodate a variety of attack conformations through dynamic binding, which may be crucial for the formation of catalytically competent complexes on different surface structures (Fig. 3D). Based on the above results, we reasoned that TurboPETase is probably more promiscuous with respect to the conformation of the PET strand it attacks. Nevertheless, detailed analysis of the mechanism requires further efforts through more in-depth research.

## Performance of TurboPETase on alternative substrates and in dual-enzyme systems

Previous study has demonstrated limited biodegradation efficiency of *Is*PETase and its variant towards other semiaromatic polyesters, specifically PBT, at 37 °C[36]. Compared to PET, PBT has a lower glass transition temperature ($T_g$), which ranges between 37 and 55 °C[48]. In this study, even though 65 °C surpasses the $T_g$ of PBT, thus significantly enhancing the mobility of PBT polymer chains, all of the examined enzymes exhibited substantially reduced degradation efficiency towards PBT films at 65 °C with respect to the degradation of PET (Supplementary Fig. 15). Specifically, TurboPETase yielded higher amounts of hydrolytic products (62.6 µM) than both BhrPETase (27.5 µM) and LCC$^{ICCG}$ (47.1 µM). These results suggested that the active sites of current PET-degrading enzymes were less efficient in binding with the extended aliphatic chains in PBT compared to PET. Consequently, dedicated efforts in enzyme discovery or tailored engineering are still needed for further improving the depolymerization of new classes of semiaromatic polyesters.

Various efforts have been made to develop dual enzyme systems to remove the intermediate products BHET and MHET[49–53], which are known inhibitors of PET hydrolases. Haugwitz et al. reported a dual enzyme system combining an engineered TfCa mutant and PETase PM enzyme that yielded a 4-fold increase in overall products towards Gf-PET films compared to PETase PM alone at 45 °C over 24 h[52]. We further explored the application potential of TurboPETase by coupling it with the recently reported thermostable BHET hydrolyzing enzyme, BHETase[54], in a dual-enzyme system. At a low substrate loading (2 g kg$^{-1}$), the dual-enzyme system effectively doubled the overall yield of the products, relative to the singular use of TurboPETase (Supplementary Fig. 16). However, an intriguing observation emerged at an elevated PET loading of 30 g kg$^{-1}$. TurboPETase alone surpassed the yields from most enzyme ratios in the dual-enzyme system at 65 °C, the only exception being the 0.5 mg$_{TurboPETase}$/g$_{PET}$:0.1 mg$_{BHETase}$/g$_{PET}$ ratio. Nevertheless, in the current investigation, only degradation of amorphous PET materials at 65 °C were evaluated. Future research could expand to examine the efficacy of the dual-enzyme system on different MHET hydrolases and PET substrates with varying physical properties under diverse reaction conditions, including varied temperatures.

## Nearly complete degradation of postconsumer PET products at industrially relevant high-solids loading

To better evaluate the depolymerization performance of TurboPETase at industrially relevant levels of solids loading, we have compared the depolymerization of pretreated postconsumer coloured-flake PET (PcPET) wastes with TurboPETase and LCC$^{ICCG}$ at a substrate loading of 200 g kg$^{-1}$. TurboPETase achieved nearly complete depolymerization (98.2%, calculated from the HPLC data) of PcPET wastes in 8 h (Fig. 4A), with a maximum production rate of 61.3 g$_{hydrolyzed\ PET}$ L$^{-1}$ h$^{-1}$ (Supplementary Fig. 17 and Supplementary Tables 9 and 10). A slight change in the crystallinity of PcPET was observed (reaching 11.9% after 8 h), which allows an immediate reuse of the remaining PcPET waste for depolymerization. In contrast, LCC$^{ICCG}$ required 16 h to reach 97.7% depolymerization (calculated from the HPLC data) at 65 °C, and the remaining PcPET exhibited a similar crystallinity of 11.1%, demonstrating that the "physical aging" process was suppressed at lower temperatures. At the previously reported optimal reaction temperature of 72 °C, LCC$^{ICCG}$ reached its maximal conversion of 92.5% (calculated from the HPLC data) over 12 h, and no further increase was obtained after prolonged reaction time due to the higher deformability of PET chains. The remaining PcPET showed a high level of

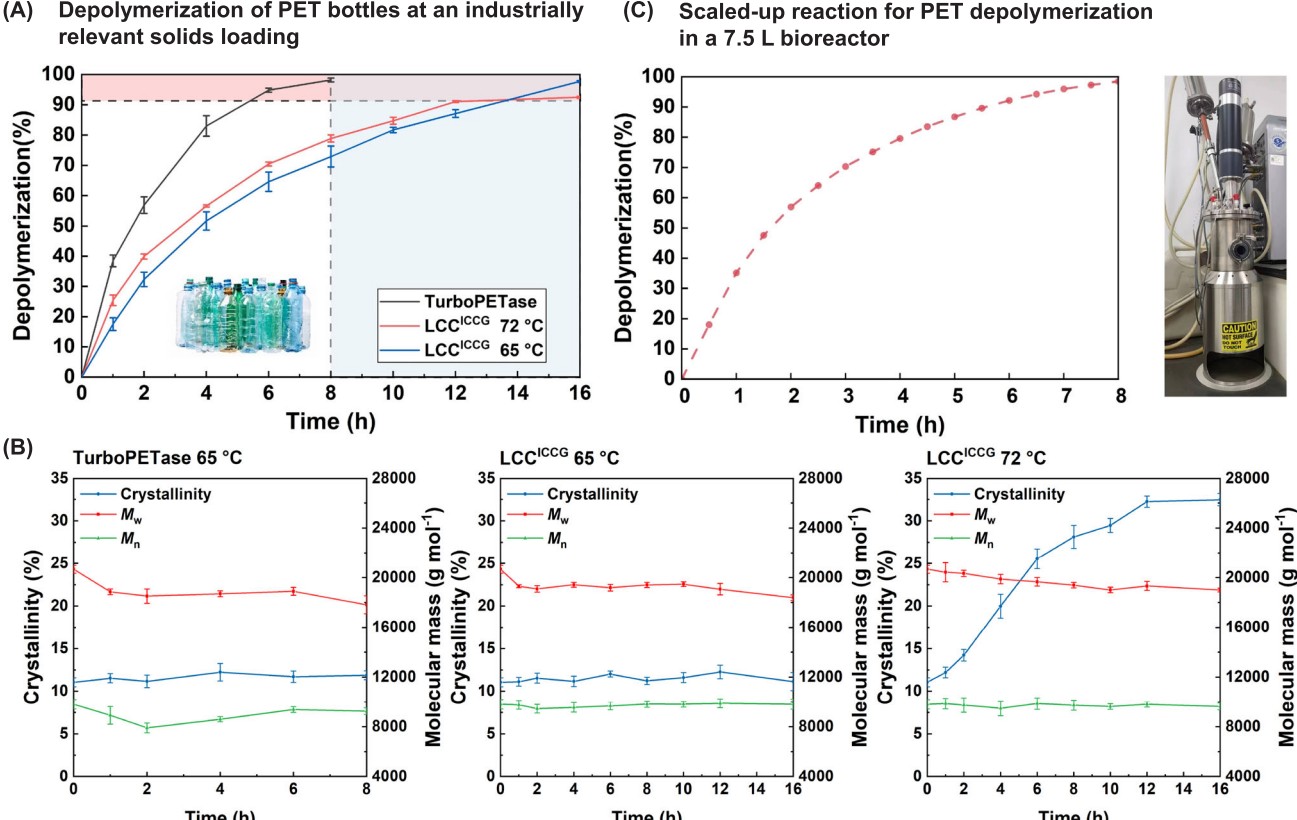

**Fig. 4 | Depolymerization of PcPET with TurboPETase and LCC$^{ICCG}$ in bioreactors. A** Comparison of PcPET depolymerization kinetics for TurboPETae at 65 °C and LCC$^{ICCG}$ at 65 °C and 72 °C. Reactions were performed with 200 g kg$^{-1}$ substrate loading and 2 mg$_{enzyme}$ g$_{PET}^{-1}$ enzyme loading at pH 8.0. PET-depolymerization percentages were calculated based on the released products measured by HPLC. Data are presented as mean ± s.d. ($n$ = 3 biologically independent experiments). **B** Time courses of crystallinity, $M_w$ and $M_n$ changes during the enzymatic hydrolysis. Data are presented as mean ± s.d. ($n$ = 3 biologically independent experiments). **C** Nearly complete degradation of pretreated PcPET bottles by TurboPETase at 65 °C with 200 g kg$^{-1}$ substrate loading and 2 mg$_{enzyme}$ g$_{PET}^{-1}$ enzyme loading in water solution in a 7.5 L bioreactor. PET-depolymerization percentages were calculated based on NaOH consumption (pH of the reaction mixture was regulated at 8.0). Data are from one independent experiment. Source data are provided as a Source Data file.

crystallinity, estimated at 32.5%, which is in accordance with a previous report[17].

Many factors can affect enzymatic attack against the plastics: e.g., crystallinity, chain mobility, molecular size, surface topography and hydrophobicity[11]. Among these, the role of low crystallinity in PET degradation has been extensively studied. A crystallinity exceeding 20% has been previously proposed to significantly impede the enzymatic degradation process. In the depolymerization of non-melt-quenched PET powders (27.6% crystallinity), we found a significant reduction in degradation performance (Supplementary Fig. 18 and Supplementary Table 11). However, during the depolymerization of pretreated PcPET, even when the crystallinity reached 20% after 4 h of degradation by LCC$^{ICCG}$ at 72 °C, we did not observe a rapid decline in degradation rate. Pfaff et al. showed that LCC$^{ICCG}$ is more efficient in hydrolyzing shorter polymers[27]. Recent studies also demonstrated that heavily pretreated PET, with a degree of polymerization (DP) less than 20 and high crystallinity, can remain highly degradable[55,56]. These findings underscored the importance of factors other than crystallinity in enzymatic degradation, and promoted further investigation through gel permeation chromatography (GPC) analysis, which revealed notably low weight-average ($M_w$) and number-average ($M_n$) molecular masses in PcPET powders compared to non-melt-quenched PET powders (Supplementary Table 12). From the GPC results, we speculate that the low molecular weight of pretreated PcPET may contribute to the maintained degradation rate by LCC$^{ICCG}$ at 72 °C during the first several reaction hours, but a further increase in crystallinity to 32.5% led to a substantial reduction in the amorphous

regions, impeding further enzymatic catalysis. Moreover, during the depolymerization process, we also observed a slight decrease in $M_w$ and moderate changes in $M_n$, despite achieving over 90% depolymerization (Fig. 4B). This is consistent with previous studies on PET degradation hydrolyzed by Cut190[57]. It has been suggested that the enzymatic hydrolysis of PET typically initiates with an endo-type chain scission after the enzyme binds to the PET surface. Further hydrolysis of the neighboring ester bonds within the amorphous regions occurs through an exo-type chain scission, leading to the release of soluble products[19]. For polymers undergoing random scission, the probability of exo-scission inversely correlates with chain length[53]. For PET with a DP around 200, the probability for exo-scission is substantially less than for endo-scission. In this study, the DP of the pretreated PcPET, approximately 50, is much lower than that of typical raw PET materials. This resulted in a significantly increased number of chain ends available for exo-scissions. Consequently, endo-type scission, which would reduce the DP of PET, no longer held a dominant proportion, which may potentially explain the minimal changes in molecular size observed during the PET depolymerization process.

To explore the industrial viability, we scaled up the reaction and performed it in a 7.5 L bioreactor with a PET loading of 200 g kg$^{-1}$ in water solution at pH 8.0. Enzyme loading is of particular concern to the industry since enzyme cost is among the essential factors in the biomass depolymerization process. In contrast to using high enzyme concentrations up to 20 mg$_{enzyme}$ g$_{PET}^{-1}$ in the biomass conversion process[13,58], 2 mg$_{enzyme}$ g$_{PET}^{-1}$ enzyme loading is saturated in PET degradation. Based on the TEA model[14], a change from

0.1 $mg_{enzyme}$ $g_{PET}^{-1}$ to 2 $mg_{enzyme}$ $g_{PET}^{-1}$ enzyme loading levels only results in a 2% increase in MSP (US$0.04/kg), whereas the capital and operating costs would increase by at least 5% to maintain the prolonged reaction duration when low enzyme loading levels are used. More importantly, in large-scale reactions with >150 g kg$^{-1}$ substrate loading, the reduction in enzyme activity and depolymerization efficiency was more significant at lower enzyme loading levels[13]. In PET degradation process reported by Tournier et al., a final conversion of 80% was achieved by LCC$^{ICCG}$ using 1 $mg_{enzyme}$ $g_{PET}^{-1}$ enzyme loading compared to the 90% conversion with 2 $mg_{enzyme}$ $g_{PET}^{-1}$ enzyme loading[17]. Hence, a 2 $mg_{enzyme}$ $g_{PET}^{-1}$ enzyme loading was chosen to attain a balance between productivity and enzyme cost. As shown in Fig. 4C, the scaled-up reaction progression is almost linear for the first 2 h with approximately 57% depolymerization achieved, and followed by a slower phase from 2 to 8 h. According to the kinetic parameters obtained from Fig. 3, we suggested that the reduced reaction rate may be attributed to the substantial decrease in PET loading at high conversion levels. Despite the decreased hydrolytic efficiency, the approximately 98% depolymerization (98.9% calculated from the HPLC data, 98.4% calculated from the consumed NaOH and 97.4% calculated from the weight loss, as listed in Supplementary Table 10) achieved within 8 h during the scaled-up reaction makes pilot-scale production feasible. Further process refinement could potentially optimize the depolymerization efficiency for lower concentrations of PET waste.

## Discussion

Biocatalytic PET depolymerization offers a sustainable and energy-efficient approach to PET recycling, presenting a more environmentally friendly alternative to current disposal methods such as landfills and incineration. While significant progress has been made to this end, the ultimate goal is to develop enzymes and processes suitable for industrial-scale applications. Both high substrate solids content and conversion benefit the economics of PET biodegradation, as it reduces both capital and operational expenditures. However, the material slurry exhibits a high apparent viscosity due to the high solids loading, leading to limited mass and heat transfer, which reduces the efficiency of enzymes in the early stages of hydrolysis. More importantly, increasing the solids loading to industrially relevant levels would lower the depolymerization yield due to the inhibition by high product concentrations[16]. Hence, mere thermostability of the enzyme may not suffice for industrial PET degradation. Multiple factors interplay, influencing enzyme efficacy in real-world scenarios. An aspect often overlooked by the scientific community is the variance in heterogeneous enzymatic hydrolysis between laboratory experiments and industrial production. For instance, the depolymerization activity of PET can be enhanced in the laboratory by fusion with noncatalytic binding modules to increase the enzyme concentration on the PET surface. However, this advantage is completely lost at an industrial solids loading level[15]. Despite recent advancements in engineering PET hydrolases to improve their performance in PET depolymerization, efforts are still needed in the quest of new PET hydrolases to address conversion loss under industrially relevant conditions.

Here, we employed a hybrid computational strategy to redesign a PET hydrolase that significantly outperforms other well-known PET hydrolases. Nearly full degradation of postconsumer PET bottles was achieved at an industrially relevant level of solids loading, rendering this highly efficient, optimized enzyme a good candidate for future applications in industrial plastic recycling processes. The mechanism responsible for enhancing enzyme performance has been demonstrated via kinetic analyses derived from an inverse Michaelis–Menten reaction regime as well as structural analysis, highlighting the importance of improving the specific polymer interactions on specific attack sites rather than general nonspecific surface adsorption. The results may help further knowledge on heterogeneous reaction catalysis and shall be beneficial for designing industrial viable plastic-degrading

enzymes to address the challenges associated with other more abundant plastics, such as polyurethanes with hydrolysable backbones. While the potential of enzyme design drives further developments in the improvement of enzyme performance, it is crucial to assess their depolymerization efficiency under industrially relevant conditions to demonstrate practical feasibility.

## Methods

### Training the Transformer model

**Unsupervised training datasets.** Homologous sequences of BhrPETase and LCC$^{ICCG}$ were searched from Uniclust30 (version 2018_08) and the BFD database with HHblits (the number of iterations was set as 4, and other parameters were left as default values) using 15 seed sequences in Pfam family PF01083 as queries. All searched sequences were clustered at 90% identity using CD-HIT to obtain 10847 sequences. Since BFD and Uniclust30 were clustered at very low similarity, they may have been undersampled for fitness modelling in very close regions. Previous work by Frazer et al.[59] sampled MSA built with more similar proteins to predict disease variants. We also retrieved 15051 sequences belonging to PF01083 from the UniProt database.

**Training details and the prediction of "less-fitted" candidates.** To model the fitness distance of sequences, we trained a neural network with an encoder-decoder from starch. We used a Transformer encoder to process input amino acid sequences with absolute position embedding. Unlike other models such as recurrent and convolutional networks, the Transformer made no assumption on sequence ordering and was more powerful at capturing long-distance relationships in sequence because of the attention mechanism (Eq. 1):

$$Attention(Q,K,V) = softmax\left(\frac{QK^T}{\sqrt{d_k}}\right)V \qquad (1)$$

We applied multi-head self-attention as described by ref. 60. The encoder consists of 3 Transformer layers with 8 heads using an embedding size of 512. Based on the encoder embeddings, the decoder generates probabilities of each token. The model was trained with a masked language modelling objective to predict the real amino acid at the masked position. In this study, 40% of tokens were replaced with mask tokens during training. We used the Adam optimizer with a learning rate set to 3 e$^{-4}$. Models were trained for 20 epochs using a batch size of 32. Residues were filtered by sorting by the logits assigned to the WT amino acid. The top ten residue positions with the highest average scores from the prediction of each model were selected. The average score was calculated using the following equation:

$$\bar{L}_{residue} = \frac{\sum_1^{19}(L_{mi} - L_{wt})}{19} \qquad (2)$$

where $\bar{L}_{residue}$ is the average score of the predicted residue position, $L_{mi}$ and $L_{wt}$ are the predicted logits of the single point mutation in the residue position and the wild-type amino acid, respectively. Excluding the duplicated positions, a total of 18 residue positions were generated, among which W104, H164, M166, W190, H191, H218, and F/I243 were suggested to be located on the PET-binding groove.

### Design of stabilizing mutations by the GRAPE strategy

The GRAPE strategy reported in our previous study[36] was used to improve the protein stability. The sequence of BhrPETase$^{H218S/F222I}$ was submitted to the GRAPE-WEB online server (https://nmdc.cn/grape-web/). Based on the sequence information of BhrPETase$^{H218S/F222I}$, AlphaFold2[47] was used to predict the structure model. Subsequently, energy calculations with FoldX[37], Rosetta_cartesian_ddg[38], and ABACUS[39] were used to predict stabilizing mutations. The DDD algorithm[40] was used to predict the

suitable disulfide bonds. The thresholds for ABACUS, FoldX, and Rosetta were set to -3.0 AEU, −1.5 kcal/mol and −1.5 REU, respectively.

## Molecular docking and MD simulations

TurboPETase was modelled by AlphaFold2. Then, the protein was simulated for 20 ns. The representative structure obtained from the MD simulations and the crystal structure of BhrPETase (PDB ID: 7EOA) was used for further docking. A model substrate comprising three consecutive PET units, mimicking the N-terminal and C-terminal groups to cap the polymer chain at both ends, was generated. YASARA was employed to make molecular docking of PET to Turbo-PETase and BhrPETase. The highest binding energy model was selected. The selected model was subjected to local docking for 999 runs. The optimized model of TurboPETase-PET and BhrPETase-PET complexes were simulated in AMBER 16[61] using the ff16SB force field. The His242 residue was protonated in the HID state. Cl⁻ ions were added to maintain system neutrality. The systems were then solvated in a truncated octahedron box using the TIP3P water model with an 8 Å distance around the solute. Following solvation, the systems underwent a 12,000-step minimization and were gradually heated from 0 K to 338 K with a collision frequency of $1 \, ps^{-1}$. After equilibration, 100 ns MD simulations were conducted for each complex, employing a time step of 2 fs.

## Cloning

Genes encoding BhrPETase (GenBank accession number: GBD22443), LCC (GenBank accession number: AEV21261), $LCC^{ICCG}$ ($LCC^{F243I/D238C/S283C/Y127G}$), $ICCG^{I6M}$ ($ICCG^{S32L/D18T/S98R/T157P/E173Q/N213P}$), FastPETase ($IsPETase^{D186H/R280A/N233K/R224Q/S121E}$, IsPETase GenBank accession number: BBYR01000074), HotPETase ($IsPETase^{S121E/D186H/R280A/N233C/S282C/P181V/S207R/S214Y/Q119K/S213E/R90T/Q182M/N212K/R224L/S58A/S61V/K95N/M154G/N241C/K252M/T270Q}$), DepoPETase ($IsPETase^{T88I/D186H/D220N/N233K/N246D/R260Y/S290P}$), PES-H1$^{L92F/Q94Y}$ and $CaPETase^{M9}$ ($CaPETase^{L180C/A202C/R242C/S291C/V129T/R198K/N109A/A155R/G196T}$, CaPETase GenBank accession number: SHM40309) were synthesized and optimized for expression in Escherichia coli (General Biosystems, Anhui, China). The signal peptide of BhrPETase, LCC, $LCC^{ICCG}$, $ICCG^{I6M}$ and $CaPETase^{M9}$ were removed from the synthetic DNA. The synthesized genes for BhrPETase, LCC, $LCC^{ICCG}$, PES-H1$^{L92F/Q94Y}$ and FastPETase were cloned into the NheI and XhoI sites of the pBAD vector (containing an N-terminal His-tag), whereas the gene for $ICCG^{I6M}$, DepoPETase, $CaPETase^{M9}$ and HotPETase were cloned into the NdeI and XhoI sites of the pET-21a(+) vector (containing a C-terminal His-tag). A list of nucleotide sequences is provided in Supplementary Table 13.

## Site-directed mutagenesis

BhrPETase mutants were generated using the QuickChange site-directed mutagenesis kit (Agilent Technologies, Santa Clara, CA, USA). The PCR products were subsequently treated with DpnI (New England Biolabs, Ipswich, MA, USA) to digest the original DNA template and then introduced into E. coli TOP10 cells. Verification of the introduced mutations was carried out through DNA sequencing (Tianyi Huiyuan, Beijing, China).

## Protein purification

Plasmids containing the genes of BhrPETase, LCC, $LCC^{ICCG}$, PES-H1$^{L92F/Q94Y}$, FastPETase and their variants were transformed into E. coli BW25113. Plasmids of $ICCG^{I6M}$, DepoPETase and $CaPETase^{M9}$ were transformed into E. coli C41(DE3), while the plasmid of HotPETase was transformed into E. coli Rosetta gami-B cells (competent cells of the expression strains were purchased from Zoman Biotech, Beijing, China). The cells were cultured in 2×YT medium at 37 °C to an $OD_{600 \, nm}$ of ~0.8, and then protein expression was induced by adding 0.2% ($w/v$) L-arabinose (BhrPETase, LCC, $LCC^{ICCG}$, PES-H1$^{L92F/Q94Y}$, FastPETase and their variants) or 1 mM isopropyl β-D-thiogalactopyranoside ($ICCG^{I6M}$, DepoPETase, $CaPETase^{M9}$ and HotPETase). The cells were cultured for 20 h at 20 °C, harvested by

centrifugation (10,000 × g, 10 min, 4 °C) and suspended in lysis buffer (50 mM $Na_2HPO_4$, 100 mM NaCl and 20 mM imidazole, pH 7.5). Cell disruption was carried out through ultrasonication on ice. The resulting cell extracts were obtained after centrifugation at 14,000 × g for 60 min at 4 °C, followed by filtration through a 0.22 μm Millex filter to remove precipitates. The unbound proteins were washed away in a 5 mL HisTrap HP column with washing buffer (50 mM $Na_2HPO_4$, 100 mM NaCl, and 60 mM imidazole, pH 7.5), the target protein was eluted with an elution buffer (50 mM $Na_2HPO_4$, 100 mM NaCl and 300 mM imidazole, pH 7.5). Afterwards, the buffer was exchanged for a storage buffer (50 mM $Na_2HPO_4$ and 100 mM NaCl, pH 7.5) by using a HiPrep 26/10 Desalting column. The purified enzyme was stored at 4 °C. Protein concentrations were determined by the BCA method with bovine serum albumin as the reference.

## PET depolymerization assay using Gf-PET films

**Evaluating the activity of the mutants during BhrPETase engineering.** The amorphous Gf-PET film (Goodfellow, 250 μm thickness, product number ES301445, ⌀8 mm, approximately 15 mg) was soaked in 500 μL of 1 M potassium phosphate buffer (pH 8.0, 30 g kg⁻¹ solids loading) with 30 μg of purified enzyme (2 $mg_{enzyme}$ $g_{PET}^{-1}$ enzyme loading) at 65 °C for 3 h. In HPLC analysis, the mobile phase consisted of 70% buffer A and 30% buffer B (Buffer A: 0.1% formic acid in distilled water; Buffer B: acetonitrile). The flow rate was 0.8 mL/min and the separation was carried out at 25 °C with detection performed at 260 nm.

**Comparing TurboPETase with other PET hydrolases under identical reaction conditions.** The initial activities of the PET hydrolases at different temperatures were estimated. The Gf-PET film (⌀8 mm, approximately 15 mg) was soaked in 500 μL of 100 mM potassium phosphate buffer (pH 8.0, 30 g kg⁻¹ solids loading) with 4.5 μg (0.3 $mg_{enzyme}$ $g_{PET}^{-1}$ enzyme loading) or 30 μg (2 $mg_{enzyme}$ $g_{PET}^{-1}$ enzyme loading) of purified enzyme at 50, 60 or 65 °C for 3 h, and then the supernatant was analysed by HPLC to quantify the concentration of released PET monomers.

Due to the high extent of conversion after prolonged reactions, the time-course analysis was performed in 1 M potassium phosphate buffer (pH 8.0) to provide sufficient buffering capacity. The Gf-PET film (⌀8 mm, approximately 15 mg) was soaked in 500 μL of 1 M potassium phosphate buffer (pH 8.0, 30 g kg⁻¹ solids loading) with 4.5 μg (0.3 $mg_{enzyme}$ $g_{PET}^{-1}$ enzyme loading) of purified enzyme at 50, 60 or 65 °C for 12 h.

**Comparing TurboPETase with other PET hydrolases under their reported reaction conditions.** The Gf-PET film (⌀8 mm, approximately 15 mg) was soaked in 750 μL buffer (20 g kg⁻¹ solids loading) for 12 h. For comparison with FastPETase, the Gf-PET film was soaked in 100 mM potassium phosphate buffer (pH 8.0) at 50 °C with 4.5 μg of purified enzyme (0.3 $mg_{enzyme}$ $g_{PET}^{-1}$ enzyme loading). For comparison with HotPETase, the Gf-PET film was soaked in 50 mM Glycine-NaOH buffer (pH 9.2) at 60 °C with 0.9 μg of purified enzyme (0.06 $mg_{enzyme}$ $g_{PET}^{-1}$ enzyme loading). For comparison with $LCC^{ICCG}$, the Gf-PET film was soaked in 100 mM potassium phosphate buffer (pH 8.0) at 65 °C with 3 μg of purified enzyme (0.2 $mg_{enzyme}$ $g_{PET}^{-1}$ enzyme loading) for 3 h.

## Depolymerization of the pretreated PET bottles

**PET bottles pretreatment.** Postconsumer PET bottles were collected from the garbage collection station in Beijing, China. The bottles were washed with deionized water and dried and then micronized into small flakes using a crusher JZ-T-005 (Shiyan Precision Instruments, Dongguan, China). The flakes were subsequently amorphized using a twin-screw extruder SY-6219-20/32 (Shiyan Precision Instruments, Dongguan, China). The set temperatures were 265 °C in the extruder zones,

285 °C in the melt pump, and 285 °C in the screen changer zones. The obtained amorphous fibres of PET were further micronized by a DFY−1000D grinder (DingLi, Wenzhou, China) at room temperature. After sieving, PET powders with particle sizes less than 400 μm were obtained (11.1% crystallinity on average).

**Comparing the depolymerization performance of TurboPETase and LCC^ICCG.** Twenty grams of pretreated PET powder and 80 mL of 100 mM potassium phosphate buffer (pH 8.0) containing 40 mg of purified enzyme were mixed in a 200 mL bioreactor (Kusnc, Shanghai, China). A magnetic stirrer was utilized to maintain constant agitation at 300 rpm. The temperature was regulated at 65 °C (or 72 °C for LCC^ICCG) by water bath immersion, and the pH was regulated at 8.0 by adding 4 M NaOH solution. Assays were performed in triplicate and evaluated accordingly.

During the reaction, the percentage of PET depolymerization was evaluated by analysing the supernatant through HPLC. The final depolymerization yield was additionally verified by investigating the weight of residual PET. The reaction mixture was filtered through Whatman grade 1 qualitative filter papers (GE Healthcare, USA). The residue was washed twice with deionized water and then dried at 65 °C overnight. To evaluate the proportion of PET in the residue, 200 mg of the dried residue was incubated in 20 mL of 4 M NaOH solution at 90 °C for 3 h. Afterwards, the amount of released TPA was measured by HPLC analysis. The overall weight of the residue solids and the proportion of PET are shown in Supplementary Table 9.

**Estimating the depolymerization performance of TurboPETase at a larger scale.** A total of 0.5 kg pretreated PET powder, 1.9 L of deionized water and 0.1 L of TurboPETase solution (containing 1 g of purified enzyme) were combined in a 7.5 L bioreactor (New Brunswick Scientific, Edison, NJ, USA). The stirring rate was maintained at 300 rpm, the temperature was regulated at 65 °C, and the pH was regulated at 8.0 by adding 4 M NaOH solution. The percentage of PET depolymerization during the reaction was evaluated according to base consumption. Additionally, the final depolymerization yield was verified by analysing the residue (alkaline hydrolysis and HPLC analysis methods were identical to the protocol above), which was collected by centrifugation (14,000 × g, 20 min), washed twice with deionized water and dried at 65 °C overnight.

**Determination of apparent melting temperatures, PET crystallinity and molecular weights**
To determine apparent melting temperatures, a fluorescence-based thermal stability assay was employed. A 5 μL 100-fold diluted SYPRO Orange dye (Molecular Probes, Life Technologies, USA) was added in a 20 μL protein solution. The mixture was placed in a thin-walled 96-well PCR plate, sealed with optical-quality sealing tape. The mixture was heated in a CFX 96 real-time polymerase chain reaction (PCR) system (BioRad, Hercules, CA, USA) from 25 to 100 °C at a heating rate of 1.4 °C/min. The crystallinity of PET and PBT was analysed by differential scanning calorimetry (DSC). The parameters were set following a previously reported procedure[36]. For molecular weight analysis of PET, gel permeation chromatography (GPC) analysis was applied. GPC measurements were carried out using an Agilent 1260 system (Agilent 1260, Agilent Technologies Inc., U.S.A.) with a Refractive Index (RI) detector. The mobile phase consisted of 1,1,1,3,3,3-hexafluoro-2-propanol (HFIP), and the analysis was performed at 35 °C with a flow rate of 1 mL/min.

**Kinetics analysis**
For PET depolymerisation, initial rate measurements were collected for the ^convMM and ^invMM datasets. For the ^convMM dataset, Gf-PET films over a range of 0-50 g kg^−1 were treated in 300 μL of 100 mM potassium phosphate buffer (pH 8.0) with enzyme loadings of 0.12 μM,

0.37 μM and 0.74 μM (1–6 μg). For the ^invMM dataset, 12 g kg^−1, 20 g kg^−1, and 30 g kg^−1 Gf-PET films were treated in 500 μL of 100 mM potassium phosphate buffer (pH 8.0) with enzyme loading levels ranging from 0 to 2.67 μM (0–36 μg). The reaction mixture was incubated at 65 °C for 1 h. All reactions were performed in triplicate and terminated by heating to 100 °C for 10 min. The supernatant obtained by centrifugation (18,000 × g, 5 min) was then analysed by HPLC. The data were fitted using Matplotlib in python.

For 4-Nitrophenol butyrate (pNPB), all reactions were performed in 96-well plates in a total reaction volume of 100 μL. The final concentrations of pNPB range from 0.2 to 1.4 mM, and the final concentration of enzymes was 0.5 μg mL^−1. 10 μL of the solution of pNPB (in anhydrous ethanol), 80 μL of 10 mM potassium phosphate buffer (pH 8.0), and 10 μL of enzymes were incubated at 65 °C for 3 min. Then, 100 μL of anhydrous ethanol was added to quench the enzymatic reactions. Each reaction was conducted in triplicate. The p-nitrophenol products were measured at 405 nm using an Infinite® 200 PRO microplate reader (Tecan, Switzerland). One unit of enzyme activity was defined as the amount of enzyme required to convert 1 μmol of pNPB per min. The data were fitted using Matplotlib in python.

For MHET, the reactions were performed in a total volume of 250 μL. The final concentrations of MHET range from 6 to 60 mM (in 100 mM potassium phosphate buffer, pH 8.0), and the final concentration of enzymes was 0.1 mg mL^−1. The reactions were conducted at 65 °C in triplicate for 15 min, then were quenched by adding 250 μL of dimethylsulfoxide. The hydrolysis extent was measured by HPLC. The data were fitted using Matplotlib in python.

**Binding isotherms**
Adsorption measurements for TurboPETase, BhrPETase, LCC^ICCG and bovine serum albumin (BSA) were conducted in a 100 mM potassium phosphate buffer (pH 8.0) using low-binding Eppendorf tubes[46]. A Gf-PET film (⌀8 mm, approximately 15 mg) was soaked in 300 μL of buffers containing protein concentrations ranging from 0 to 2 μM, maintained at 65 °C for 1 h equilibration period. Afterward, 100 μL of the supernatant was mixed with 100 μL of freshly prepared BCA working solution. This mixture underwent incubation at 37 °C for 30 min, followed by absorbance measurements at 562 nm using a plate reader to determine the concentration of free proteins within the solution. The quantities of bound proteins were then deduced by calculating the difference between the total protein concentrations and the concentrations of free proteins.

**Determination of free cysteine residues**
To identify the free cysteine residues in the solution of TurboPETase, 5,5'-Dithiobis-(2-nitrobenzoic acid) (DTNB) was used[27]. 25 μL of enzyme solution (8 μM of BhrPETase or TurboPETase) or L-cysteine solution (0, 10, 20, 30, 40 and 50 μM), 225 μL of freshly prepared DTNB solution (2 mg/mL DTNB in 100 mM potassium phosphate buffer, pH 8.0) and 250 μL of EDTA solution (100 mM ethylene diamine tetraacetic acid in 100 mM potassium phosphate buffer, pH 8.0) were mixed thoroughly and incubated for 15 min. Afterward, the absorption of samples at 412 nm was measured using the spectrophotometer (MAPADA V−1100D, Shanghai, China).

**Depolymerization of PBT**
The PBT films were soaked in 500 μL of 100 mM potassium phosphate buffer (pH 8.0, 4 g kg^−1 solids loading) with 4 μg of purified enzyme (2 mg_enzyme g_PBT^−1 enzyme loading) at 65 °C for 3 h.

**Depolymerization with a dual-enzyme system**
The Gf-PET film (⌀3 mm, approximately 2 mg) was soaked in 1 mL of 100 mM potassium phosphate buffer (pH 8.0, 2 g kg^−1 solids loading) containing a two-enzyme system (2 mg_TurboPETase g_PET^−1 TurboPETase

and BHETase loading ranging between 0 and 2 mg$_{BHETase}$ g$_{PET}$$^{-1}$) and incubated at 65 °C for 9 h. For a substrate loading of 30 g kg$^{-1}$, the Gf-PET film (⌀8 mm, approximately 15 mg) was soaked in 500 μL of 100 mM potassium phosphate buffer (pH 8.0, 30 g kg$^{-1}$ solids loading) containing the two-enzyme system. The mixture was incubated at 65 °C for 1 h with TurboPETase and BHETase concentrations ranged from 0 to 2 mg$_{enzyme}$ g$_{PET}$$^{-1}$. The supernatant was analysed by HPLC.

## Reporting summary

Further information on research design is available in the Nature Portfolio Reporting Summary linked to this article.

## Data availability

All data that support the findings of this study are presented within the article and its supplementary files. For further inquiries or requests for additional information, please contact the corresponding authors. Source data are provided with this paper.

## Code availability

The Transformer model used in this study has been made publicly available at https://github.com/Wublab/code-for-TurboPETase.

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

## Acknowledgements

This work was supported by the National Key R&D Program of China (grant no. 2021YFC2103600 to C.L.L.), the National Natural Science Foundation of China (32225002 to B.W., 31822002 to B.W. and 32170033 to Y.L.C.), the Key Research Program of Frontier Sciences (ZDBS-LY-SM014 to B.W.), the Biological Resources Program (KFJ-BRP-009 to B.W. and KFJ-BRP-017-58 to Y.L.C.) from the Chinese Academy of Sciences, the Informatization Plan of Chinese Academy of Sciences (CAS-WX2021SF-0111 to B.W.), the Youth Innovation Promotion Association CAS (2022086 to Y.L.C.), and the Postdoctoral Fellowship Program of CPSF (GZC20232929 to T.Z.).

## Author contributions

B.W. initiated the project. Y.L.C. and J.Y.S. performed the computational work, Y.C.C., H.P., and C.L.L. performed biochemical and biocatalytic experiments, T.Z. and W.C.G. performed polymer degradation experiments, Y.L.C. and B.W. drafted the manuscript, which was revised and approved by all authors.

## Competing interests

The authors declare no competing interests.
