## [Peer Review File · Nature Communications]

Computational redesign of a hydrolase for nearly complete PET depolymerization at industrially relevant high-solids loadingREVIEWER COMMENTS

Reviewer #1 (Remarks to the Author):

Wu and coworkers present an interesting study about optimising the enzyme PETase for PET plastic recycling. The optimized PETase shows superior performance compared to previously PETase mutants in industrially-relevant conditions.

The work is important as PET plastic accumulation in the environment is a global problem and current recycling techniques are unsatisfactory.

The research is competently done, and the results are mainly supported by the data, even though not entirely.

Nevertheless, there are two specific aspects that make me to advise against publication at this stage, which I detail below:

1. Even though turboPETase outperforms previous PETase mutants, the increase in performance is not very substantial. I was not convinced that turboPETase is a breakthrough for plastic recycling. It represents a step forward on a mainly incremental work of enzyme optimization and not a novel solution or a radically improved solution. In this sense, I consider that the work does not have the level of novelty and relevance necessary for publishing in Nat. Methods.

2. The work is more technical than scientific. The authors develop a tool for recycling PET, which they show that works well, but don't explain why it works well. The proposed increased flexibility of the active site may turn true. Still, more profound and solid studies are needed to explain the origin and rationale of the increase in enzyme performance. Such rationale is what distinguishes high-level science from technical work because it makes us understand better how nature works and inspires others in the field about ways of reaching similar goals, supporting the process of the scientific field in general. Therefore, I advise the authors to focus more seriously on explaining why the enzyme performance increased and present solid proof for their hypotheses. Otherwise the work leads to technical progress but not scientific advances.

Reviewer #2 (Remarks to the Author):

This manuscript authored by Cui et al. was previously submitted to [another Nature journal], and I had the privilege of serving as a reviewer during that time. It appears that the authors have subsequently resubmitted their original manuscript to Nature Communications without making any revisions based on the feedback received from the various reviewers including myself. Consequently, I believe it is reasonable to expect that the authors should address the comments I provided during its initial submission to [another Nature journal] before I can proceed with any further recommendations. My previous review report is presented below.

The authors have made improvements to their manuscript by replacing certain previous data (in the main manuscript in their initial submission to Nature), particularly concerning the inconsistent use of buffer and temperature conditions. This revision allows for a more impartial comparison against BhrPETase (wild-type enzyme), FastPETase, HotPETase, and LCC-ICCG, considering reaction temperatures up to 65°C and reaction durations up to 3 hours. Within the scope of investigating higher solid loadings at larger scales, the authors eliminated a number of less promising data, including the use of PET waste with higher crystallinity, which resulted in slower conversion rates and lower efficiency at 65°C with TurboPETase (partially also tested under alternative buffer conditions). I have reservations about the complete removal of these data, as it prevents interesting scientific discussions of the impact of various PET waste sources and their thermal history under pretreatment, and critical buffer conditions on the degradation performance under industrially-relevant conditions. In addition to retaining one old dataset using TurboPETase and the most heavily pretreated waste PET (with a resulting crystallinity of 10.9%), the authors have also included new datasets using the reference benchmark enzyme LCC ICCG. These new datasets demonstrate that TurboPETase revealed significantly superior degradation performance at 65°C compared to LCC ICCG at both 65°C and 72°C (although there are concerns about the data's reliability at the latter condition, as detailed in the comments below). The achievement of complete depolymerization in an 8-hour buffer-free system using TurboPETase, which has been successfully upscaled to a 7.5 L bioreactor, is impressive and

suggests the potential of TurboPETase as an industrially applicable biocatalyst for PET recycling. Despite these remarkable accomplishments and improvements (such as additional data and clearer descriptions), this revised manuscript does not fundamentally transform its original characteristics as incremental research on "computational simulation/structural biology-based engineering" of yet-another "thermophilic PETase." It follows a similar workflow to the work published by Tournier et al. (2020) in Nature and numerous other similarly structured papers published in other high-impact journals since then. In addition to these questions about novelty, I have further inquiries and doubts that should be addressed by incorporating new data and discussions, regardless of whether this manuscript is considered for publication in this target journal or elsewhere.

1.) Because the wt BhrPETase is 94% identical to the wt LCC, the resulting mutant TurboPETase with 8 single mutations is still very similar to the reference LCC ICCG enzyme. Since a wider audience may not be aware of this high similarity due to the nomenclatures used to call these enzymes, which is also not emphasized in the revised manuscript, I believe it is reasonable to include a figure in the main text (e.g., an extended form of Fig. S5 to include structural and sequence information also about LCC and ICCG variant instead of only being shown in the supplementary information) to illustrate the sequence alignment of these four enzyme variants as well as a structural alignment based on co-crystallized ligand (e.g., PDB ID 7VVE) to highlight the variable positions that have been mutated in this study. Possibly also the few additional ones that were left out of the mutagenesis but are still close to the binding groove.

2.) Although the authors could have revised the rationale for using the language model and other computational tools (yellow-highlighted text on page 3 and the supplementary discussion on page 11 of the supplementary information file), the utility of these models appears to be quite limited to merely identifying two mutagenesis target positions which are obviously not unique: W104 has been reported by Zeng et al. (ref. no. 30 regarding protein engineering of the highly similar LCC ICCG benchmark enzyme) for its role in the interaction with the substrate; F243 (and its equivalent position) has been repeatedly identified as a mutagenesis hot spot for LCC ICCG (e.g., by Tournier et al. in Nature and many follow-up researches published afterwards) or other highly homologous PETases (e.g., ref. no. 32). Therefore, the novelty, usefulness and effectiveness of this method should not be overemphasized as described in the current version of the manuscript, e.g., the statement in the rebuttal letter "The results showed that W104 and F243 are difficult to predict by simple sequences alignment, demonstrating the advantages of using machine learning to uncover hidden information regarding the improvement of polymer degradation." Accordingly, the limitation of this method should be stated.

In contrast, the GRAPE strategy, despite its limited novelty, has suggested four of the eight useful residue substitutions in TurboPETase, which appears to be more effective for universal use. In this context, as described in lines 267 and 268, the A251C/A281C double mutations should form a disulfide bridge that is distinct from those introduced to LCC-ICCG and many other PETases at an alternative position. I am unable to find experimental evidence that the A251C/A281C disulfide bridge is actually formed. This must be demonstrated using structural biology techniques or biochemical methods.

3.) In Figure 2, the authors could present new experimental data for comparing different PETases under optimal buffer and temperature conditions. In general, lower ionic strengths and buffer concentrations are used in the new experiments, which are also related to the final depolymerization experiments with high substrate solid loading. Nonetheless, based on the information provided in the supplementary information, 1.7(1) and 1.10, the comparison of the activity of various mutants (Fig. 1D) and kinetic analysis (Figure 3 and Table 1) still only included data collected with 1 M potassium buffer, which can significantly (rather positively) influence enzyme activity, stability, and adsorption behavior to the substrate (, and this is widely accepted by scientific communities). This may render the mutant ranking invalid and result in incomparable kinetic properties to the degradation experiments depicted in Figures 2 and 4. The ideal solution is to repeat the kinetic analysis at lower buffer concentrations and, at the very least, probe several mutants shown in Fig. 1D to ensure that the use of 1 M buffer has no (negative) effect on their ranking in enzyme activity comparison. Furthermore, neither the main text nor the supplementary information explains how and in which buffer the T_m of individual mutants was determined. As previously stated, if 1 M buffer was used, certain mutants could be sufficiently thermoactivated and stabilized. Therefore, the superior mutant TurboPETase deduced based on the data in Fig. 1D and the supplementary tables may not be the ideal one to be used in a buffer-free condition (Fig. 4B).

Comparing the old datasets (which have been moved to Fig. S4) with the new datasets in Fig. 2, it was also observed that the use of 1 M buffer can improve the degradation performance of TurboPETase by approximately 4-fold (at 50°C) or 7-fold (at 65°C) at similar enzyme to substrate

ratios but low solid loading levels. From an opposing perspective, have the authors considered confirming this benefit of using a high buffer concentration for large-scale, high-solid-loading experiments such as those depicted in Figure 4? For a future industrial application in the real world, a high buffer concentration will undoubtedly raise numerous (such as cost-related) questions. However, it would be of great scientific interest to determine if a 1 M buffer will further increase the degradation efficiency of TurboPETase and, if so, to discuss whether it would be preferable to use a low salt concentration at the expense of drastically reducing the degradation efficiency.

4.) The depolymerization performance with TurboPETase, as shown in Figure 4, and the maximum production rate described in line 289, are unquestionably the highlights of this study. However, based on the polymer property data available thus far, the PET substrate used in this study has a crystallinity of 10.9%, which is significantly lower than the 14.6% used by Tournier et al (2020) with LCC ICCG. The final crystallinity of residual PET materials at 65°C as a result of physical aging and degradation-induced crystallization is in the range of 12.3-12.8%, which is still significantly lower than the starting material used by Tournier et al. As a result, I believe that the remarkable degradation performance should rather be attributed to the more "heavily amorphized" waste PET in comparison to the less amorphized materials used by Tournier et al. Furthermore, ref. 32. indicated that the molecular weights of the pretreated PET waste may influence their degradability; thus, the authors should provide GPC analysis of their pristine PET waste, those after extrusion, after micronization, and after enzymatic degradation individually for a better understanding of the correlation between polymer property and degradability.

5.) The inferior performance of LCC ICCG at 65°C shown in Fig. 4 appears plausible to me, consistent with the data shown in Fig. 2 and also by Tournier et al., who stated that LCC ICCG should perform similarly to wt LCC at 65°C. However, the marginally better-performed degradation curve with LCC ICCG at 72°C with a similar shape to that determined at 65°C until at least 14 h did not appear to be reasonable. This is in stark contrast to previous studies such as those by Tournier et al., Zeng et al. (ref. no. 30), and this recently published one by Ding et al. (<https://doi.org/10.1016/j.jhazmat.2023.131386>) also focusing on further protein engineering of LCC ICCG. All previous studies clearly demonstrated a significantly higher (~1.5 to 2-fold) degradation activity of LCC ICCG at temperatures ranging from 72-74°C compared to those at 65°C to 66°C. Given the lower crystallinity of the substrate used in this study (10.9%), the degradation performance with LCC ICCG at 72°C did not appear to be reliable or reproducible (i.e. should be much higher). Furthermore, as shown in Fig. S1B, the crystallinity of the PET substrate at 72°C exceeded 20% after 4 hours of reaction; however, it did not appear that the LCC-ICCG reaction rates were significantly reduced before 12 hours. The degradation curve at 72°C behaved in some ways like a "parallel" curve with the one determined at 65°C, where the crystallinity of the PET substrate continuously maintained less than 15% in correlation to a high degradability. This makes absolutely no sense to me.

6.) I observed that the degradation curves in Fig. 4A and Fig. S6 were calculated using HPLC analysis, whereas the one depicted in Fig. 4B was determined using only NaOH consumption. I believe that these data ought to be better validated, similar to those presented by Tournier et al. in their extended Fig. 5 to calculate normalized depolymerization curves based on combinatorial data obtained by NaOH consumption, HPLC analysis of the formation of TPA/MHET/BHET; EG, and the weight loss determinations, with the original data presented separately in a table.

7.) I am unable to fully comprehend the statement between lines 299 and 301, which I find to be quite confusing. I believe it is necessary for the authors to provide data and more thorough descriptions of the "water constraint" and "supersaturated TPA" concentration profiles under various reaction conditions.

8.) Statements in lines 20-21 and 356-357 appear to be exaggerated. At least in refs. 30 and 32, as well as the omitted publication by Ding et al. (see above for DOI), additional enzymes or mutants that outperform LCC-ICCG under specific reaction conditions, have been demonstrated. Obviously, these enzymes are not included in this study for experimental comparison with TurboPETase, nor are they even given serious consideration. As a highly active research field, additional promising enzyme variants may be published during this manuscript's revision stage. Before overselling their own findings, I strongly advise the authors to remain open to learning about the most recent developments in this research field.

Reviewer #3 (Remarks to the Author):

The manuscript entitled Computational redesign of a hydrolase for nearly complete PET depolymerization at industrially relevant high-solids loading provides some interesting information related to enzyme development. However, given the high number of articles published in this and other journals related to incremental improvements of PET hydrolyzing activity of various enzymes, the novelty of this article seems to be very limited. The authors tried to compare the activity of their enzyme to some recently published PET hydrolyzing enzymes while it is not clear how e.g. different reaction conditions or enzyme thermostabilities may affect the comparison in figure 4. Apart from decomposition of PET, a lot of other parameters would be interesting to be included in a more mechanistic comparison of different PET hydrolyzing enzymes, such as sorption characteristics, activity on small water soluble (model) substrates, release of oligomers and activity thereon, specificity for other aromatic / aliphatic polyesters, potential synergies with other enzymes etc.

General responses to the reviewers' comments

We sincerely appreciate the careful way the reviewers examined the manuscript and are pleased with the positive comments. Meanwhile, their constructive criticisms and valuable suggestions are of great help in refining the manuscript. Accordingly, extensive additional experiments were performed and the manuscript has been substantially revised. Here's a brief overview of the key revisions:

- (1) **Adsorption experiments:** We have added the adsorption experiments utilizing a simple Langmuir approach, offering deeper insights into the binding efficiency of TurboPETase to heterogeneous PET films. Steady-state kinetics on soluble substrates (MHET and *p*NPB) were also performed to allow for a more comprehensive discussion on the potential reasons for enhanced depolymerization capabilities of TurboPETase.
- (2) **Structural analysis:** In the revised manuscript, we have expanded our analysis to the potential local structural changes induced by the mutations. The new insights may offer a more nuanced molecular understanding of TurboPETase's catalytic attributes.
- (3) **Hydrolysis efficiency under the request conditions:** We introduced new kinetic analyses and tested a suite of 12 mutants based on the M6 scaffold under conditions using a 100 mM KH_2PO_4 -NaOH buffer. Additionally, we evaluated the depolymerization of TurboPETase and LCC^{ICCG} on high-crystallinity PET materials to address the concern raised by reviewer #2 about whether TurboPETase is only efficient on extremely low-crystallinity PET substrates.
- (4) **Extended applications:** To explore the potential broader applications of TurboPETase, we assessed its synergistic effects with BHETase. Additionally, we conducted supplementary experiments on the depolymerization of another aromatic polyester, PBT.
- (5) **Comparative analysis:** During our submission process, a series of works emerged on the engineering of PET-degrading enzymes. We selected the most recent and influential enzymes for further comparative analysis, with TurboPETase still outpacing. Nonetheless, we agree with the reviewer's comments and have revised our manuscript to "TurboPETase is more superior compared to the well-known enzymes."

We deeply appreciate the reviewers' feedback, which prompted us to undertake supplementary experiments that substantially enhance the robustness and reliability of our study. The precise amendments are elaborated upon below and are highlighted in yellow in the revised manuscript.

Point-to-point responses to the reviewers' comments:

Reviewer 1:

Comment 1: 1. Even though turboPETase outperforms previous PETase mutants, the increase in performance is not very substantial. I was not convinced that turboPETase is a breakthrough for plastic recycling. It represents a step forward on a mainly incremental work of enzyme optimization and not a novel solution or a radically improved solution. In this sense, I consider that the work does not have the level of novelty and relevance necessary for publishing in *Nat. Methods*.

Response: We genuinely appreciate the reviewer's acknowledgment of the merits in our work. Over the last 3 years, we have witnessed a number of success in the engineering of PET hydrolases, gearing towards potential applications in industrial plastic recycling processes. However, a majority of works aiming at enhancing enzyme performance are performed in small-scale reactions, often at

low enzyme and substrate concentrations, even though large-scale reaction experiments up to 200 g kg⁻¹ substrate loading have been reported (*Nature* 2020, 580, 216). In heterogeneous catalysis reactions, different enzymes may exhibit dramatically distinct catalytic efficiencies at various enzyme and substrate concentrations, as in the case of cellulases (*ACS Catal.* 2017, 7, 4904-4914; *Biotechnol Biofuels.* 2020, 13, 58). For instance, Erickson *et al.* demonstrated that *IsPETase* showed similar hydrolytic conversion compared to its variant *IsPETase*^{W159H/S238F} at low enzyme loading (0.5-1 mg_{enzyme} g_{PET}⁻¹), but approximately 2-fold lower conversion at higher enzyme loading (0.5-1 mg_{enzyme} g_{PET}⁻¹). More importantly, the enzyme activity would be severely reduced at high product concentrations in an industrially relevant scale. Alongside the solids increase, a decrease in the final yield is often observed in enzymatic hydrolysis of biomass, which is generally known as the “high-solids effect”. The overall conversion of cellulose decreased from ~65% to 40% as solids loadings increased from <100 g kg⁻¹ to 175 g kg⁻¹ (*Biomass. Bioenergy* 2013, 56, 526-544). This has been inconclusively attributed to water constraint or the inhibition of enzymes by the high concentration of its products (*Biotechnol Biofuels.* 2020, 13, 58).

In our comparative analysis with the well-known engineered PET-degrading enzymes, their enzymatic activity under industrially-relevant conditions (high substrate loadings, elevated enzyme concentrations) failed to exceed that of LCC^{ICCG}, although they demonstrated remarkable degradation capabilities in their reported conditions (Figure 2). Hence, despite the progressive strides in PET enzyme engineering, the efficiency of PET enzymatic hydrolysis under industrial conditions remains stagnant.

Concurring with the reviewer's astute observation, we recognize that this manuscript could benefit from an infusion of more scientifically robust content. We have added a more in-depth analysis of the structural features and the heterogeneous catalytic properties of TurboPETase that may explain its improved performance, as well as analysis of the catalytic performance of TurboPETase for different substrates (high crystallinity PET, soluble small molecule substrates). We sincerely hope that our revised manuscript could meet the publication standards of **Nature Communications**. Most importantly, we wish to convey to the wider bio-scientific community the significance of assessing enzymatic depolymerization efficiency under conditions pertinent to industry, ensuring the real-world applicability of subsequent enzyme engineering initiatives.

Comment 2: The work is more technical than scientific. The authors develop a tool for recycling PET, which they show that works well, but don't explain why it works well. The proposed increased flexibility of the active site may turn true. Still, more profound and solid studies are needed to explain the origin and rationale of the increase in enzyme performance. Such rationale is what distinguishes high-level science from technical work because it makes us understand better how nature works and inspires others in the field about ways of reaching similar goals, supporting the process of the scientific field in general. Therefore, I advise the authors to focus more seriously on explaining why the enzyme performance increased and present solid proof for their hypotheses. Otherwise the work leads to technical progress but not scientific advances.

Response: Thanks for the valuable comment. In our revised manuscript, we incorporated a series of new experiments to elucidate the potential factors contributing to the enhanced performance of TurboPETase. Firstly, we conducted a kinetic comparison for soluble substrates (MHET and *p*NPB). The catalytic efficiency of TurboPETase towards MHET showed only a modest enhancement relative to PET (with a 32% increase). This observation prompted us to consider alternative factors,

possibly an increased adsorption to the surface, as a potential contribution to the amplified degradation efficiency of PET. We subsequently measured free enzyme concentrations, E_{free} , and converted it to substrate coverage, $\Gamma = (E_{tot} - E_{free})/S_{PET}$ to calculate the adsorption of enzymes to PET surface. Intriguingly, upon reaching saturation, the enzymes, including TurboPETase, manifested similar maximum adsorption capacities (Γ_{max}), underscoring a consistent overall binding potential across the enzymes. Since non-specific adsorption accounts for a considerable proportion of the total adsorption sites, comparisons of these enzymes could be expanded by the consideration of inverse Michaelis–Menten parameters, which can reflect the binding capability to the specific attack sites of the PET surface. The $^{inv}K_M$ values revealed marginal differences between TurboPETase, BhrPETase, and LCC^{ICCG}, whereas the $^{inv}V_{max}/^{mass}S_0$ of TurboPETase exhibited a 2.3-fold enhancement. Given that no substantial differences in $^{inv}K_M$ and Γ_{max} values were observed among the enzymes, the elevated $^{inv}V_{max}/^{mass}S_0$ values may imply a broadened targeting of TurboPETase towards specific attack sites when the enzymes maintained a stable overall adsorption level. Detailed discussion can be found in page 7-8, lines 233-276, “Additionally, we conducted a kinetic comparison for soluble substrates (MHET and *p*NPB) as detailed in Supplementary Fig.S7 and Supplementary Table S6. The catalytic efficiency of TurboPETase towards MHET showed only a modest enhancement relative to PET (with a 32% increase), suggesting that there may be other factors, potentially increased adsorption to the surface, contributing to the amplified degradation efficiency of PET. Conversely, the slight decrement in TurboPETase’s k_{cat} for *p*NPB, accompanied by a reduced binding affinity, inferred potential changes in the substrate binding domain, rendering it less conducive for other small molecule interactions.

Although the hydrolysis of PET could not meet the criteria for the conventional approach, the inverse Michaelis–Menten model was more applicable... It should be noted that not all adsorption sites are competent for catalytic conversion, non-specific adsorption also accounts for a considerable proportion⁴³. We measured free enzyme concentrations, E_{free} , and converted it to substrate coverage, $\Gamma = (E_{tot} - E_{free})/S_{PET}$ to calculate the total adsorption of the enzymes to PET surface (Supplementary Fig. S8 and Supplementary Table S7)... Consequently, we presumed that the enhanced depolymerization performance of TurboPETase may rely, at least in part, on the enhanced ability to attack a broader spectrum of specific attack sites that can be hydrolysed to form a productive complex.”

Upon analyzing the Michaelis–Menten parameters for *p*NPB, we found a slight decrease in TurboPETase’s k_{cat} accompanied by a reduced binding affinity. This suggests potential changes in the substrate binding domain of TurboPETase, rendering it less conducive for other small molecule interactions. Therefore, we have provided a more detailed analysis of potential local structural changes and impacts on PET binding induced by the mutations. The new insights may offer a more nuanced molecular understanding of the enhanced catalytic performance of TurboPETase. Please refer to page 10-11, lines 305-329, “H218 is suggested to form an intimate packing with the conserved W190 in analogous enzymes. Chen *et al.* found that PET hydrolytic activity could benefit from a more flexible active site in the H214S/F218I double mutant (corresponds to H218S/F222I in BhrPETase)³⁵. In the present study, MD simulations of the apo form of TurboPETase revealed an expanded rotational freedom of W190 endowed by the H218S/F222I mutation (Supplementary Fig. S11), which is consistent with the observation of diverse conformations of the corresponding W156 of *Is*PETase³⁵. When binding to the PET, the wobbling of W190 is curtailed and anchored by the π - π interactions with the PET substrate. This flexibility of the PET binding cleft was further enhanced by the synergistic interactions conferred by the addition of W104L and F243T, as revealed by the C α root-mean-square fluctuation (RMSF) results (Fig. 3C). W104 is previously reported to pack against the adjacent P248 to stabilize the P248-situated β 8- α 6 loop³⁴. In the redesigned TurboPETase, the relinquishment of this interaction by leucine

substitution may engender increased conformational malleability within the loop region (N246-A250), as demonstrated by the largely reduced cross-correlation of these regions (Supplementary Fig. S12). For another substitution F243T in the PET binding cleft, the steric profile of F243 appears to dictate a more peripheral binding locus for PET. Yet, its mutation to threonine, armed with a less pronounced steric feature, may release the space for PET binding with a more-flexible state. More importantly, without the steric profile of the aromatic ring, T243 may beckon PET deeper into the cleft, drawing the substrate's labile carbonyl closer to the catalytic serine, with the interstitial distances contracting from $4.88 \pm 0.51 \text{ \AA}$ to $4.15 \pm 0.37 \text{ \AA}$ (Supplementary Fig. S13). Concurrently, the enhanced flexibility might compromise the protein's stability, which is consistent with the observed decrease in the melting temperatures of the single point mutations.”

Synthesizing our structural analysis with the kinetic data, we postulated that the greatly increased flexibility along the PET-binding groove may provide more space to accommodate a variety of attack conformations through dynamic binding to enhance the ability to attack a broader spectrum of specific attack sites. We earnestly hope that our revised manuscript offers more insights into the potential mechanisms underlying the improved depolymerization performance of TurboPETase, and now strikes a balance between technical progression and scientific advancement.

Reviewer 2:

Comment 1: Because the wt BhrPETase is 94% identical to the wt LCC, the resulting mutant TurboPETase with 8 single mutations is still very similar to the reference LCC ICCG enzyme. Since a wider audience may not be aware of this high similarity due to the nomenclatures used to call these enzymes, which is also not emphasized in the revised manuscript, I believe it is reasonable to include a figure in the main text (e.g., an extended form of Fig. S5 to include structural and sequence information also about LCC and ICCG variant instead of only being shown in the supplementary information) to illustrate the sequence alignment of these four enzyme variants as well as a structural alignment based on co-crystallized ligand (e.g., PDB ID 7VVE) to highlight the variable positions that have been mutated in this study. Possibly also the few additional ones that were left out of the mutagenesis but are still close to the binding groove.

Response: Thanks for the reviewer's comment. We recognize the importance of providing the sequence information for these enzymes, as it would offer readers a clearer insight into their interrelationships. In the revised manuscript, we have added the structural comparison of TurboPETase with its counterparts LCC^{ICCG} and BhrPETase as depicted in Figure 2. The local structure of the mutation site has been magnified to ensure readers can distinctly perceive the location of the mutation. Given the limitations of incorporating extensive content into the main figure, we added the sequence alignment of these enzymes to the Supplementary information. Additionally, we've elucidated the relationship between these enzymes. Kindly refer to page 3, lines 97-100 in the revised manuscript, “BhrPETase shares a high sequence identity of 94% with LCC, whereas LCC^{ICCG} represents a variant of LCC characterized by four amino acid substituents (F243I/D238C/S283C/Y127G). Both FastPETase and HotPETase are derivatives engineered from *Is*PETase (Supplementary Figure S3).”

Comment 2: Although the authors could have revised the rationale for using the language model and other computational tools (yellow-highlighted text on page 3 and the supplementary discussion on page 11 of the supplementary information file), the utility of these models appears to be quite limited to merely identifying two mutagenesis target positions which are obviously not unique: W104 has been reported by Zeng et al. (ref. no. 30 regarding protein engineering of the highly

similar LCC ICCG benchmark enzyme) for its role in the interaction with the substrate; F243 (and its equivalent position) has been repeatedly identified as a mutagenesis hot spot for LCC ICCG (e.g., by Tournier et al. in *Nature* and many follow-up researches published afterwards) or other highly homologous PETases (e.g., ref. no. 32). Therefore, the novelty, usefulness and effectiveness of this method should not be overemphasized as described in the current version of the manuscript, e.g., the statement in the rebuttal letter “The results showed that W104 and F243 are difficult to predict by simple sequences alignment, demonstrating the advantages of using machine learning to uncover hidden information regarding the improvement of polymer degradation.” Accordingly, the limitation of this method should be stated.

In contrast, the GRAPE strategy, despite its limited novelty, has suggested four of the eight useful residue substitutions in TurboPETase, which appears to be more effective for universal use. In this context, as described in lines 267 and 268, the A251C/A281C double mutations should form a disulfide bridge that is distinct from those introduced to LCC-ICCG and many other PETases at an alternative position. I am unable to find experimental evidence that the A251C/A281C disulfide bridge is actually formed. This must be demonstrated using structural biology techniques or biochemical methods.

Response: Thanks for the reviewer’s comment. Indeed, as noted by the reviewer, position F243 represents a mutated site in LCC^{ICCG} as reported by Tournier et al. However, W104 position has not been reported to enhance the depolymerization performance so far. In reference 30, Zeng et al. suggested that W104 has packing interactions with its mutated residue N248P thereby enhancing the protein thermostability, rather than an interaction with the substrate to promote its depolymerization properties. Nonetheless, we understand the reviewer’s concerns about the usefulness and effectiveness of the Transformer model. We have revised our statement regarding this method, discussing the limitations and potential future optimization directions for this approach. Please refer to the revised Supplementary Discussion, “These results demonstrated the difficulty in predicting W104 and F243 using bioinformatics methods. Recent successes in enzyme design guided by statistical models or neural networks informed by protein family data or multiple sequence alignment (MSA) have highlighted the rich information encoded in the sequence space of natural enzymes associated with certain functionalities¹¹⁻¹³. While the Transformer model offers some promising insights, it is not devoid of limitations. One notable concern is its potential inadequacy for orphan enzymes that lack a substantial number of homologous sequences. Secondly, it is challenging to distinguish functionally relevant signals from noise in diverse sequences. Given the increased variability in the N-terminal and C-terminal sequences compared to the core of the protein, our method had a bias toward more variable at N- and C- terminals. Manual removal of such regions is necessary. Additionally, the current model does not consider the structural information, which has been demonstrated as valuable in other engineering efforts¹⁴⁻¹⁵. Incorporating such information presents a potential avenue for refining our approach. Furthermore, given the extensive research efforts directed towards the engineering of PET hydrolases, integrating available experimental measurement data could further enhance the robustness and accuracy of the algorithm in subsequent iterations.”

For disulfide bond, we have confirmed the disulfide bond formation using Ellman’s reagent, which has also been used by Pfaff et al. (*ACS Catal.* 2022, 12, 9790-9800). Please refer to Supplementary Figure S10.

Comment 3: In Figure 2, the authors could present new experimental data for comparing different PETases under optimal buffer and temperature conditions. In general, lower ionic strengths and buffer concentrations are used in the new experiments, which are also related to the final depolymerization experiments with high substrate solid loading. Nonetheless, based on the

information provided in the supplementary information, 1.7(1) and 1.10, the comparison of the activity of various mutants (Fig. 1D) and kinetic analysis (Figure 3 and Table 1) still only included data collected with 1 M potassium buffer, which can significantly (rather positively) influence enzyme activity, stability, and adsorption behavior to the substrate (, and this is widely accepted by scientific communities). This may render the mutant ranking invalid and result in incomparable kinetic properties to the degradation experiments depicted in Figures 2 and 4. The ideal solution is to repeat the kinetic analysis at lower buffer concentrations and, at the very least, probe several mutants shown in Fig. 1D to ensure that the use of 1 M buffer has no (negative) effect on their ranking in enzyme activity comparison.

Furthermore, neither the main text nor the supplementary information explains how and in which buffer the T_m of individual mutants was determined. As previously stated, if 1 M buffer was used, certain mutants could be sufficiently thermoactivated and stabilized. Therefore, the superior mutant TurboPETase deduced based on the data in Fig. 1D and the supplementary tables may not be the ideal one to be used in a buffer-free condition (Fig. 4B).

Comparing the old datasets (which have been moved to Fig. S4) with the new datasets in Fig. 2, it was also observed that the use of 1 M buffer can improve the degradation performance of TurboPETase by approximately 4-fold (at 50°C) or 7-fold (at 65°C) at similar enzyme to substrate ratios but low solid loading levels. From an opposing perspective, have the authors considered confirming this benefit of using a high buffer concentration for large-scale, high-solid-loading experiments such as those depicted in Figure 4? For a future industrial application in the real world, a high buffer concentration will undoubtedly raise numerous (such as cost-related) questions. However, it would be of great scientific interest to determine if a 1 M buffer will further increase the degradation efficiency of TurboPETase and, if so, to discuss whether it would be preferable to use a low salt concentration at the expense of drastically reducing the degradation efficiency.

Response: Thanks for the reviewer's comment. The T_m was determined in the enzyme storage buffer (50 mM Na₂HPO₄ and 100 mM NaCl, pH 7.5). We have added the descriptions in the methods section in the supplementary materials. Please refer to section "1.9 Determination of apparent melting temperatures and PET crystallinity" in the revised Supplementary materials, "A fluorescence-based thermal stability assay was used to determine apparent melting temperatures. Protein solution (20 μ L) in the buffer (50 mM Na₂HPO₄, 100 mM NaCl, pH 7.5) was mixed with 5 μ L 100-fold diluted SYPRO Orange dye (Molecular Probes, Life Technologies, USA) in a thin-walled 96-well PCR plate. The plate was sealed with optical-quality sealing tape and heated in a CFX 96 real-time polymerase chain reaction (PCR) system (BioRad, Hercules, CA, USA) from 25 to 100 °C at a heating rate of 1.4 °C/min. Fluorescence changes were monitored with a charge-coupled device (CCD) camera. The wavelengths for excitation and emission were 490 and 575 nm, respectively. The parameters were set following a previously reported procedure³."

As the reviewer suggests, we compared the depolymerization performance of the 12 variants based on the M6 scaffold (Figure R1) and executed additional experiments for kinetic analysis employing 100 mM KH₂PO₄-NaOH buffer. As shown in Figure R1, TurboPETase was still the most active variant in respect to other variants at lower buffer concentrations.

Figure R1. Comparison of the PET-hydrolytic activity of the M6 variants towards Gf-PET films in different buffer concentrations. Reactions were performed at 65 °C using 30 g kg⁻¹ solids loading and 2 mg_{enzyme} g_{PET}⁻¹ enzyme loading for 3 hours.

The new kinetic results also demonstrated comparable $^{inv}K_m$ among the evaluated enzymes and improved $^{inv}V_{max}/^{mass}S_0$ values for TurboPETase, which are similar to the results in the original manuscript. Thus, the main conclusion has not changed. We have redrawn the figures with the new data, please refer to Figures 3A and B in the revised manuscript.

Furthermore, we agree with the reviewer that higher phosphate concentrations (up to 1 M) may largely promote the PET hydrolysis activity for certain enzymes, such as PES-H1 (*ACS Catal.* 2022, 12, 9790-9800) and M6-W104G/F243T mutant in this study. For TurboPETase, however, the performance enhancement under these conditions appears to be more restrained. A 1M phosphate concentration could improve the hydrolysis activity by a modest ~10% at an enzyme loading of 2 mg_{enzyme} g_{PET}⁻¹ at 65 °C. This increment is slightly more pronounced, around 34%, at lower enzyme concentrations (0.3 mg_{enzyme} g_{PET}⁻¹) at 65 °C. We speculate that the 4-fold (at 50°C) or 7-fold (at 65°C) increase inferred by the reviewer may stem from the comparison of degradation performance under varying enzyme concentrations and substrate loadings. Additionally, the disparity could also be attributed to the comparison with LCC. The activity of LCC is inhibited under conditions with 1 M phosphate buffer concentration, thereby widening the disparity in catalytic efficiency when compared to TurboPETase. To facilitate a more transparent comparison, we listed the degradation efficiency of TurboPETase under different conditions in the appended table. These data can also be found in the source data submitted to both [another Nature journal] and Nature Communications. Considering the marginal influence of buffer concentration on TurboPETase’s degradation efficiency at high enzyme concentrations, coupled with the considerable economic and environmental costs associated with phosphate removal in industrial reaction conditions, we advocate for preserving the current conditions for large-scale reactions.

Table R1. PET monomers released from hydrolysing Gf-PET films with TurboPETase at temperatures ranging from 50 to 65 °C for 3 h, using solids loading of 30 g kg⁻¹. All measurements were conducted in triplicate (n = 3).

Temperature	Buffer:	Buffer:	Buffer:	Buffer:
-------------	---------	---------	---------	---------

(°C)	100 mM K-Pi Enzyme loading: 0.3 mg_{enzyme} g_{PET}⁻¹	1 M K-Pi Enzyme loading: 0.3 mg_{enzyme} g_{PET}⁻¹	100 mM K-Pi Enzyme loading: 2 mg_{enzyme} g_{PET}⁻¹	1 M K-Pi Enzyme loading: 2 mg_{enzyme} g_{PET}⁻¹
50	4.02 ± 0.14 (mM)	6.78 ± 0.83 (mM)	4.20 ± 0.34 (mM)	6.25 ± 0.11 (mM)
60	9.57 ± 0.11 (mM)	16.12 ± 0.80 (mM)	11.73 ± 0.20 (mM)	17.95 ± 0.23 (mM)
65	19.75 ± 0.72 (mM)	26.50 ± 1.16 (mM)	29.66 ± 1.25 (mM)	32.76 ± 1.58 (mM)

Comments 4-6: The depolymerization performance with TurboPETase, as shown in Figure 4, and the maximum production rate described in line 289, are unquestionably the highlights of this study. However, based on the polymer property data available thus far, the PET substrate used in this study has a crystallinity of 10.9%, which is significantly lower than the 14.6% used by Tournier et al (2020) with LCC ICCG. The final crystallinity of residual PET materials at 65°C as a result of physical aging and degradation-induced crystallization is in the range of 12.3-12.8%, which is still significantly lower than the starting material used by Tournier et al. As a result, I believe that the remarkable degradation performance should rather be attributed to the more "heavily amorphized" waste PET in comparison to the less amorphized materials used by Tournier et al. Furthermore, ref. 32. indicated that the molecular weights of the pretreated PET waste may influence their degradability; thus, the authors should provide GPC analysis of their pristine PET waste, those after extrusion, after micronization, and after enzymatic degradation individually for a better understanding of the correlation between polymer property and degradability.

The inferior performance of LCC ICCG at 65°C shown in Fig. 4 appears plausible to me, consistent with the data shown in Fig. 2 and also by Tournier et al., who stated that LCC ICCG should perform similarly to wt LCC at 65°C. However, the marginally better-performed degradation curve with LCC ICCG at 72°C with a similar shape to that determined at 65°C until at least 14 h did not appear to be reasonable. This is in stark contrast to previous studies such as those by Tournier et al., Zeng et al. (ref. no. 30), and this recently published one by Ding et al. (<https://doi.org/10.1016/j.jhazmat.2023.131386>) also focusing on further protein engineering of LCC ICCG. All previous studies clearly demonstrated a significantly higher (~1.5 to 2-fold) degradation activity of LCC ICCG at temperatures ranging from 72-74°C compared to those at 65°C to 66°C. Given the lower crystallinity of the substrate used in this study (10.9%), the degradation performance with LCC ICCG at 72°C did not appear to be reliable or reproducible (i.e. should be much higher). Furthermore, as shown in Fig. S1B, the crystallinity of the PET substrate at 72°C exceeded 20% after 4 hours of reaction; however, it did not appear that the LCC-ICCG reaction rates were significantly reduced before 12 hours. The degradation curve at 72°C behaved in some ways like a "parallel" curve with the one determined at 65°C, where the crystallinity of the PET substrate continuously maintained less than 15% in correlation to a high degradability. This makes absolutely no sense to me.

I observed that the degradation curves in Fig. 4A and Fig. S6 were calculated using HPLC analysis, whereas the one depicted in Fig. 4B was determined using only NaOH consumption. I believe that these data ought to be better validated, similar to those presented by Tournier et al. in their extended Fig. 5 to calculate normalized depolymerization curves based on combinatorial data obtained by NaOH consumption, HPLC analysis of the formation of TPA/MHET/BHET; EG, and the weight loss determinations, with the original data presented separately in a table.

Response: Thank you for the insightful comments. We deeply value the reviewer's expertise and

have taken the observations seriously, ensuring our data is not only accurate but also aligns with the broader scientific consensus.

According to the reviewer's suggestion, we have revised the depolymerization data based on the combinatorial data obtained by NaOH consumption, HPLC analysis, and the weight loss determinations. Striving for the consistency with the work of Tournier et al., we recalibrated the initial rate, specifically basing it on NaOH consumption. For a detailed data, please refer to the Supplementary Table S9. The following is the content from Supplementary Table S9.

Table S9. Enzymatic depolymerization of pretreated PET powder in the bioreactors.

Enzymatic treatment scale	Enzyme, temperature and time	Depolymerization (%) calculated from			Initial rate from consumed NaOH ^[d] (g _{hydrolyzed PET} L ⁻¹ h ⁻¹)
		Produced TPA _{eq.} ^[a]	Consumed NaOH ^[b]	Residual solids ^[c]	
20 g pretreated PET powder ^[e]	TurboPETase 65 °C, 8 h	98.2 ± 0.6	94.7 ± 3.2	96.7 ± 0.8	61.3 ± 7.0
	LCC ^{ICCG} 72 °C, 16 h	92.5 ± 0.3	86.8 ± 2.5	89.6 ± 0.3	41.2 ± 5.2
	LCC ^{ICCG} 65 °C, 16 h	97.7 ± 0.3	90.7 ± 1.6	95.5 ± 0.3	25.8 ± 2.7
500 g pretreated PET powder	TurboPETase 65 °C, 8 h	98.9	98.4	97.4	70.2

Our recalibrated initial rate for LCC^{ICCG} at 72°C, quantified as 41.2 ± 5.2 g_{TPAeq}L⁻¹h⁻¹ using NaOH consumption, aligns with the initial rate reported by Tournier et al. (40.3 g_{TPAeq}L⁻¹h⁻¹). Notably, the depolymerization values derived from NaOH consumption appeared to be somewhat lower than that calculated from HPLC, which can be attributed to the residual unhydrolyzed MHET present in the solvent. We have labeled the data source in the revised manuscript, please refer to page 11, lines 343-352, “TurboPETase achieved nearly complete depolymerization (98.2%, calculated from the HPLC data) of PpPET wastes in 8 h (Fig. 4A),...In contrast, LCC^{ICCG} required 16 hours to reach 97.7% depolymerization (calculated from the HPLC data) at 65 °C, ...At the previously reported optimal reaction temperature of 72 °C, LCC^{ICCG} reached its maximal conversion of 92.5% (calculated from the HPLC data) over 12 h, and no further increase was obtained after prolonged reaction time due to the higher deformability of PET chains.” And page 13, lines 386-388, “Despite the decreased hydrolytic efficiency, the approximately 98% depolymerization (98.9% calculated from the HPLC data, 98.4% calculated from the consumed NaOH and 97.4% calculated from the weight loss, as listed in Supplementary Table S9) achieved within 8 hours during the scaled-up reaction (Fig. 4D) makes pilot-scale production feasible.”

Upon further analysis of our recalibrated data, the initial rate for LCC^{ICCG} at 72°C was 1.6 times that at 65°C. This observation resonates with the reviewer's point and previously documented studies that underscore LCC^{ICCG}'s augmented degradation activity (by around 1.5 to 2-fold at elevated temperatures (72-74°C) compared to 65°C. Furthermore, we also extracted the remaining PET (~20% crystallinity) from reaction solvent after 4 h reaction. As per the reviewer's request, GPC measurements of the plastic were also provided (Table R2). Following thorough washing and desiccation, we added fresh LCC^{ICCG} for further degradation. The results showed a comparable

hydrolysis activity of LCC^{ICCG} on 10.7% crystallinity PET at 65°C to that on 20% crystallinity PET at 72°C (Table R3). This suggests a parallel degradation trajectory of LCC^{ICCG} at 72°C to that at 65°C between 4-8 hours of reaction. Pfaff et al. demonstrated that LCC^{ICCG} hydrolyzed shorter polymers more efficiently (*ACS Catal.* 2022, 12, 9790-9800). We speculate that due to the much lower Mn of the pretreated PcPET respect to other PET materials, LCC^{ICCG} didn't exhibit a rapid decline in degradation rate during the reaction at 72°C when the PET crystallinity reached 20%. However, when the crystallinity increased to 33%, the substantial reduction in the amorphous regions hindered further catalysis.

Table R2. Crystallinity and molecular mass of PET materials before and after pretreatments determined by DSC and GPC, respectively. Mw: weight average molecular mass, Mn: number average molecular mass.

Sample name	Crystallinity (%) by DSC	Molecular mass by GPC
Pretreated PcPET powders	10.7%	Mw:19652 Mn:9198
PcPET wastes after 4 h reaction	20%	Mw:17405 Mn:8713
Crushed PcPET powders	27.6%	Mw:59798 Mn:27233
Untreated Gf-PET film	7%	Mw:43180 Mn:20051

Table R3. PET monomers released from hydrolysing PET powders with LCC^{ICCG} at 65 °C and 72 °C for 1 h, using solids loading of 30 g kg⁻¹ and enzyme loading of 2 mg_{enzyme} g_{PET}⁻¹ in 100 mM KH₂PO₄-NaOH buffer, pH 8.0. All measurements were conducted in triplicate (n = 3).

Samples (crystallinity)	65 °C	72 °C
Pretreated PcPET powders (10.7%)	16.73 ± 0.73 (mM h ⁻¹)	24.43 ± 1.07 (mM h ⁻¹)
PcPET wastes after 4 h reaction (20%)	12.34 ± 0.35 (mM h ⁻¹)	15.74 ± 0.51 (mM h ⁻¹)
Crushed PcPET powders (27.6%)	3.29 ± 0.04 (mM h ⁻¹)	3.96 ± 0.07 (mM h ⁻¹)

We acknowledge the crystallinity discrepancy (10.9% in our study vs. 14.6% by Tournier et al.). Regrettably, employing the same techniques, we couldn't achieve samples with comparable crystallinity. To mitigate this, we assessed TurboPETase's degradation performance on readily accessible crushed PET powders sourced from Coca-Cola bottles (with a crystallinity of 27.6%), ensuring that a broad range of researchers could replicate our findings. As depicted in Figure R2, TurboPETase's degradation efficiency for high-crystallinity plastic is 1.5 times that of LCC^{ICCG} (Figure R2), demonstrating that TurboPETase's augmented catalytic prowess isn't narrowly tailored to plastics with low crystallinity but is robustly applicable to high-crystallinity PET as well.

Figure R2. PET monomers released from hydrolysing crushed PcPET powders with TurboPETase LCC^{ICCG} at 65 °C for 3 h, using solids loading of 30 g kg⁻¹ and enzyme loading of 2 mg_{enzyme} g_{PET}⁻¹ in 100 mM KH₂PO₄-NaOH buffer, pH 8.0. All measurements were conducted in triplicate (n = 3).

Comment 7: I am unable to fully comprehend the statement between lines 299 and 301, which I find to be quite confusing. I believe it is necessary for the authors to provide data and more thorough descriptions of the "water constraint" and "supersaturated TPA" concentration profiles under various reaction conditions.

Response: We apologize for the confusion caused by this description. To prevent similar queries from other readers and considering that this observation does not significantly contribute to the main narrative of the article, we have removed this statement from our original submission to Nature Communications.

Comment 8: Statements in lines 20-21 and 356-357 appear to be exaggerated. At least in refs. 30 and 32, as well as the omitted publication by Ding et al. (see above for DOI), additional enzymes or mutants that outperform LCC-ICCG under specific reaction conditions, have been demonstrated. Obviously, these enzymes are not included in this study for experimental comparison with TurboPETase, nor are they even given serious consideration. As a highly active research field, additional promising enzyme variants may be published during this manuscript's revision stage. Before overselling their own findings, I strongly advise the authors to remain open to learning about the most recent developments in this research field.

Response: Thanks for the valuable comment that leads us to perform additional experiments to achieve more comparative analysis with other benchmark PETases. The realm of PETase research is indeed dynamic, and we recognize the importance of staying abreast of the latest advancements in the field.

As suggested by the reviewer, we have incorporated a comparative analysis of TurboPETase's depolymerization efficiency against DepoPETase (*Angew. Chem. Int. Ed. Engl.* **2023**, 62(14), e202218390), CaPETase^{M9} (*Nat. Commun.* **2023**, 14, 4556), PES-H1^{L92F/Q94Y} (*ACS Catal.* **2022**, 12, 9790–9800), and ICCG^{I6M} (*J. Hazard. Mater.* **2023**, 453, 131386) across a temperature range of 50 to 65 °C under standardized conditions (30 g kg⁻¹ solids loading and 2 mg_{enzyme} g_{PET}⁻¹ enzyme loading in 100 mM KH₂PO₄-NaOH buffer, pH 8.0). Given the reported enhancements of PES-H1^{L92F/Q94Y} in 1 M KH₂PO₄-NaOH buffer, we also compared TurboPETase with PES-H1^{L92F/Q94Y} under elevated buffer concentration. Across these conditions, TurboPETase consistently manifested an enhanced PET-hydrolytic activity compared to the other PET hydrolases. Details are shown in revised Figure 2 and Supplementary Fig. S6 in the revised manuscript, and please also refer to page 6, lines 181-205, "Counterparts like BhrPETase, LCC, LCC^{ICCG}, ICCG^{I6M}, and PES-H1^{L92F/Q94Y} which exhibit

high degradation performance at elevated temperatures, rendered hydrolytic activity 1.8-, 4.7-, 2.1-, 2.0-, and 19-fold lower than that of TurboPETase, respectively. At suboptimal temperatures, TurboPETase consistently outperformed other PET hydrolases, albeit the decreased hydrolytic activity. At the optimal temperature of HotPETase and CaPETase^{M9} of 60 °C, TurboPETase generated 11.73 mM monomer product in 3 h, whereas HotPETase and CaPETase^{M9} produced 4.32 mM and 0.60 mM monomers, which were 1.7-, and 18-fold lower than that of TurboPETase. When subjected to 50 °C, TurboPETase registered hydrolytic efficiencies exceeding those of FastPETase and DepoPETase by 43% and 59%, respectively. ... In light of the lower enzyme concentrations employed in certain reports, we recalibrated our reactions to mirror these conditions to ensure a fairer comparison (Figs 2C-2E). Under their reported reaction conditions, the reaction rates of LCC^{LCCG}, HotPETase, and FastPETase were 1.8-, 4.9-, and 1.0-fold lower, respectively, than that of TurboPETase. Given the reported enhancements of PES-H1^{L92F/Q94Y} in 1 M KH₂PO₄-NaOH buffer, we also compared TurboPETase with PES-H1^{L92F/Q94Y} under elevated buffer concentration, with TurboPETase still outpacing, yielding up to 2.5 times the degradation products of PES-H1^{L92F/Q94Y} at 65 °C (Supplementary Fig. S6).” Nonetheless, we agree with the reviewer's comments and have revised our manuscript to “The redesigned variant, TurboPETase, outperformed other well-known PET hydrolases.” in the abstract and also in page 2, lines 70-78, “The redesigned variant (TurboPETase) derived from this campaign outperformed the most efficient PET hydrolases currently recognized in the field (LCC^{LCCG}, LCCICCG^{LCCG}, ICCG^{LCCG}, BhrPETase^{LCCG}, FastPETase^{LCCG}, HotPETase^{LCCG}, DepoPETase^{LCCG}, CaPETase^{M9}, and PES-H1^{L92F/Q94Y}) over a range of temperatures (50 °C-65 °C). The extraordinary degradation performance afforded by TurboPETase allowed nearly complete depolymerization of postconsumer PET bottles in 8 h at a high industrially relevant substrate loading of 200 g kg⁻¹, with a maximum production rate of 61.3 g_{hydrolyzed PET} L⁻¹ h⁻¹, addressing the challenge regarding residual nonbiodegradable PET waste.”

Once again, we extend our gratitude for your insightful comments, which have undeniably enriched the depth and rigor of our study. Specifically, your guidance has allowed our data and analyses to align more closely with the standards of our field, bolstering the quality and credibility of our work. We are truly honored to receive such constructive and professional feedback. Thank you for your substantial contribution to enhancing the quality of our paper.

Reviewer 3:

Comment 1: The manuscript entitled Computational redesign of a hydrolase for nearly complete PET depolymerization at industrially relevant high-solids loading provides some interesting information related to enzyme development. However, given the high number of articles published in this and other journals related to incremental improvements of PET hydrolyzing activity of various enzymes, the novelty of this article seems to be very limited. The authors tried to compare the activity of their enzyme to some recently published PET hydrolyzing enzymes while it is not clear how e.g. different reaction conditions or enzyme thermostabilities may affect the comparison in figure 4. Apart from decomposition of PET, a lot of other parameters would be interesting to be included in a more mechanistic comparison of different PET hydrolyzing enzymes, such as sorption characteristics, activity on small water soluble (model) substrates, release of oligomers and activity thereon, specificity for other aromatic / aliphatic polyesters, potential synergies with other enzymes etc.

Response: We appreciate the meticulous feedback provided by the reviewer, highlighting the need for a more comprehensive analysis on enzyme development for PET hydrolysis. As the reviewer

suggests, we have added a more in-depth analysis of the catalytic properties of heterogeneous systems, the catalytic performance of TurboPETase for different substrates, specificity for other aromatic polyesters (PEN and PBT), and potential synergies with the recently reported BHETase. Firstly, we conducted a kinetic comparison for soluble substrates (MHET and *p*NPB). The catalytic efficiency of TurboPETase towards MHET showed only a modest enhancement relative to PET. This observation prompted us to consider alternative factors, possibly an increased adsorption to the surface, as a potential contribution to the amplified degradation efficiency of PET. We subsequently measured free enzyme concentrations, E_{free} , and converted it to substrate coverage, $\Gamma = (E_{tot} - E_{free})/S_{PET}$ to calculate the adsorption of enzymes to PET surface. Intriguingly, upon reaching saturation, the enzymes, including TurboPETase, manifested similar maximum adsorption capacities (Γ_{max}), underscoring a consistent overall binding potential across the enzymes. Since non-specific adsorption accounts for a considerable proportion of the total adsorption sites, comparisons of these enzymes could be expanded by the consideration of inverse Michaelis–Menten parameters, which can reflect the binding capability to the specific attack sites of the PET surface. The $^{inv}K_M$ values revealed marginal differences between the enzymes, whereas the $^{inv}V_{max}/^{mass}S_0$ of TurboPETase exhibited a 2.1-fold enhancement, which may imply a broadened targeting of TurboPETase towards specific attack sites when the enzymes maintained a stable overall adsorption level. Detailed discussion can be found in page 7-8, lines 233-276, “Additionally, we conducted a kinetic comparison for soluble substrates (MHET and *p*NPB) as detailed in Supplementary Fig. S7 and Supplementary Table S6. The catalytic efficiency of TurboPETase towards MHET showed only a modest enhancement relative to PET (with a 32% increase), suggesting that there may be other factors, potentially increased adsorption to the surface, contributing to the amplified degradation efficiency of PET. Conversely, the slight decrement in TurboPETase’s k_{cat} for *p*NPB, accompanied by a reduced binding affinity, inferred potential changes in the substrate binding domain, rendering it less conducive for other small molecule interactions.”

Although the hydrolysis of PET could not meet the criteria for the conventional approach, the inverse Michaelis–Menten model was more applicable... It should be noted that not all adsorption sites are competent for catalytic conversion, non-specific adsorption also accounts for a considerable proportion⁴³. We measured free enzyme concentrations, E_{free} , and converted it to substrate coverage, $\Gamma = (E_{tot} - E_{free})/S_{PET}$ to calculate the total adsorption of the enzymes to PET surface (Supplementary Fig. S8 and Supplementary Table S7)... Consequently, we presumed that the enhanced depolymerization performance of TurboPETase may rely, at least in part, on the enhanced ability to attack a broader spectrum of specific attack sites that can be hydrolysed to form a productive complex.”

Upon analyzing the Michaelis–Menten parameters for *p*NPB, we found a slight decrease in TurboPETase’s k_{cat} accompanied by a reduced binding affinity. This suggests potential changes in the substrate binding domain of TurboPETase, rendering it less conducive for other small molecule interactions. Therefore, we have provided a more detailed analysis of potential local structural changes and impacts on PET binding induced by the mutations. The new insights may offer a more nuanced molecular understanding of the enhanced catalytic performance of TurboPETase. Please refer to page 10-11, lines 305-329, “H218 is suggested to form an intimate packing with the conserved W190 in analogous enzymes. Chen *et al.* found that PET hydrolytic activity could benefit from a more flexible active site in the H214S/F218I double mutant (corresponds to H218S/F222I in BhrPETase)³⁵. In the present study, MD simulations of the apo form of TurboPETase revealed an expanded rotational freedom of W190 endowed by the H218S/F222I mutation (Supplementary Fig. S11), which is consistent with the observation of diverse conformations of the corresponding W156 of *Is*PETase³⁵. When binding to the PET, the wobbling of W190 is curtailed and anchored

by the π - π interactions with the PET substrate. This flexibility of the PET binding cleft was further enhanced by the synergistic interactions conferred by the addition of W104L and F243T, as revealed by the C α root-mean-square fluctuation (RMSF) results (Fig. 3C). W104 is previously reported to pack against the adjacent P248 to stabilize the P248-situated β 8- α 6 loop³⁴. In the redesigned TurboPETase, the relinquishment of this interaction by leucine substitution may engender increased conformational malleability within the loop region (N246-A250), as demonstrated by the largely reduced cross-correlation of these regions (Supplementary Fig. S12). For another substitution F243T in the PET binding cleft, the steric profile of F243 appears to dictate a more peripheral binding locus for PET. Yet, its mutation to threonine, armed with a less pronounced steric feature, may release the space for PET binding with a more-flexible state. More importantly, without the steric profile of the aromatic ring, T243 may beckon PET deeper into the cleft, drawing the substrate's labile carbonyl closer to the catalytic serine, with the interstitial distances contracting from 4.88 \pm 0.51 Å to 4.15 \pm 0.37 Å (Supplementary Fig. S13). Concurrently, the enhanced flexibility might compromise the protein's stability, which is consistent with the observed decrease in the melting temperatures of the single point mutations. Synthesizing our structural analysis with the kinetic data, we postulated that the greatly increased flexibility along the PET-binding groove may provide more space to accommodate a variety of attack conformations through dynamic binding, which may be crucial for the formation of catalytically competent complexes on different surface structures (Fig. 3D).”

To extend the potential applications of TurboPETase, we also examined the use of this enzyme for the degradation of another semiaromatic polyesters, PBT (Supplementary Figure S15). Compared to PET, PBT has slightly lower strength and rigidity, slightly better impact resistance, and a slightly lower glass transition temperature. Even though 65°C surpasses the glass transition temperature of PBT (T_g ranging between 37 to 55°C), thus substantially enhancing the mobility of PBT polymer chains, all of the examined enzymes exhibited substantially reduced degradation efficiency toward PBT films at 65 °C with respect to the degradation of PET. Specifically, TurboPETase yielded higher amounts of hydrolytic products (62.6 μ M) than both BhrPETase (27.5 μ M) and LCC^{YCCG} (47.1 μ M). These results demonstrated the challenge for the active sites of current PET-degrading enzymes to efficiently binding with the extended aliphatic chains in PBT, compared to those in PET. We have added the discussion in the revised Supplementary Discussion, please refer to the section “2.2 Depolymerization of Polybutylene terephthalate (PBT)” in the Supplementary materials, “We examined the use of TurboPETase for the degradation of another semiaromatic polyester, PBT (Fig. S15). Compared to PET, PBT has slightly lower strength and rigidity, slightly better impact resistance, and a slightly lower glass transition temperature. Even though 65°C surpasses the glass transition temperature of PBT (T_g ranging between 37 to 55°C¹⁶), thus substantially enhancing the mobility of PBT polymer chains, all of the examined enzymes exhibited substantially reduced degradation efficiency toward PBT films at 65 °C with respect to the degradation of PET. ... Thus, dedicated efforts in enzyme discovery or tailored engineering are still needed for further improving the depolymerization of new classes of semiaromatic polyesters.”

We further explored the potential of TurboPETase by coupling it with the recently reported BHET hydrolyzing enzyme, BHETase (*Nat. Commun.* 2023, 14, 4169), in a dual-enzyme system. At a low substrate loading (2 g kg⁻¹), this combination substantially elevated the overall yield of the products (sum of BHET, MHET and TPA), relative to the singular use of TurboPETase. However, an intriguing observation emerged at an elevated PET loading of 30 g kg⁻¹. TurboPETase alone surpassed the yields from most enzyme ratios in the dual-enzyme system, the only exception being the 0.5 mg_{TurboPETase}/g_{PET}:0.1 mg_{BHETase}/g_{PET} ratio. This observed trend echoes a previously report wherein binding modules were added to LCC^{YCCG} (*Chem Catal.* 2022, 2, 2644–2657). Owing to the

ability of fusion enzymes to hydrolyze BHET, the resultant fusion enzymes exhibited superior performance at low substrate loadings (< 3 wt% PET). However, as the PET loading intensified (up to 10-20 wt%), the fusion enzymes TrCBM1, TtCBM10, and StCBM64 show no sustained advantage over the LCC^{YCCG} domain alone over the enzyme concentration range 50 nM to 1 mM. They proposed a potential explanation suggesting that the increased solids loading increases the frequency of enzyme-substrate interactions, thereby accelerating PET hydrolysis to such an extent that the presence of the other enzyme no longer provides any additional benefit. In the current investigation, only amorphous PET materials were evaluated. Hence, subsequent studies could explore the behavior of the dual-enzyme system across PET substrates possessing varied physical morphologies, taking into account factors like crystallinity, accessible surface area, and chemical purity. However, such an exploration falls outside the primary focus of this manuscript. We have added the discussion in the revised Supplementary Discussion, please refer to the section “2.3 Depolymerization of Gf-PET films with a dual-enzyme system” in the Supplementary materials, “We further explored the coupling of TurboPETase with the recently reported BHET hydrolyzing enzyme, BHETase¹⁷, in a dual-enzyme system. At a low substrate loading (2 g kg⁻¹), the dual-enzyme system effectively doubled the overall yield of the products (sum of BHET, MHET, and TPA), relative to the singular use of TurboPETase (Fig. S16). However, an intriguing observation emerged at an elevated PET loading of 30 g kg⁻¹. TurboPETase alone surpassed the yields from most enzyme ratios in the dual-enzyme system, the only exception being the 0.5 mg_{TurboPETase}/g_{PET}:0.1 mg_{BHETase}/g_{PET} ratio. This observed trend echoes a previously report wherein binding modules were added to LCC^{YCCG} 18. The fusion enzymes exhibited superior performance at low substrate loadings (< 3 wt% PET). However, as the PET loading intensified (up to 10-20 wt%), they show no sustained advantage over the LCC^{YCCG} domain alone over the enzyme concentration range 50 nM to 1 mM. Nonetheless, in the current investigation, only amorphous PET materials were evaluated. Additional investigations could be conducted to explore the behavior of the dual-enzyme system across PET substrates possessing varied physical morphologies, taking into account factors like crystallinity, accessible surface area, and chemical purity.”

Based on these results and the previously reported observations, we'd like to further emphasize the importance of standardizing the reaction conditions to an industrially relevant setting for comparing the performance of various enzymes.

As in the response to reviewer 1, we have witnessed a number of success in the engineering of PET hydrolases over the last 3 years, gearing towards potential applications in industrial plastic recycling processes. However, a majority of works aiming at enhancing enzyme performance are performed in small-scale reactions, often at low enzyme and substrate concentrations. In heterogeneous catalysis reactions, different enzymes may exhibit dramatically distinct catalytic efficiencies at various enzyme and substrate concentrations, as in the case of cellulases (*ACS Catal.* 2017, 7, 4904-4914; *Biotechnol Biofuels.* 2020, 13, 58). For instance, Erickson *et al.* demonstrated that IsPETase showed similar hydrolytic conversion compared to its variant IsPETase^{W159H/S238F} at low enzyme loading (0.5-1 mg_{enzyme} g_{PET}⁻¹), but approximately 2-fold lower conversion at higher enzyme loading (0.5-1 mg_{enzyme} g_{PET}⁻¹). More importantly, the enzyme activity would be severely reduced at high product concentrations in an industrially relevant scale. Alongside the solids increase, a decrease in the final yield is often observed in enzymatic hydrolysis of biomass, which is generally known as the “high-solids effect”. The overall conversion of cellulose decreased from ~65% to 40% as solids loadings increased from <100 g kg⁻¹ to 175 g kg⁻¹ (*Biomass. Bioenergy* 2013, 56, 526-544). This has been inconclusively attributed to water constraint or the inhibition of

enzymes by the high concentration of its products (*Biotechnol Biofuels*. 2020, 13, 58). In our comparative analysis with the well-known engineered PET-degrading enzymes, their enzymatic activity under industrially-relevant conditions (high substrate loadings, elevated enzyme concentrations) failed to exceed that of LCC^{ICCG}, although they demonstrated remarkable degradation capabilities in their reported conditions (Figure 2). Hence, despite the progressive strides in PET enzyme engineering, the efficiency of PET enzymatic hydrolysis under industrial conditions remains stagnant. We have added this discussion in the revised manuscript, please refer to page 13, lines 398-405, “However, the material slurry exhibits a high apparent viscosity due to the high solids loading, leading to limited mass and heat transfer, which reduces the efficiency of enzymes in the early stages of hydrolysis. More importantly, increasing the solids loading to industrially relevant levels would lower the depolymerization yield due to the inhibition by high product concentrations¹⁶. Hence, mere thermal stability of the enzyme may not suffice for industrial PET degradation. Multiple factors interplay, influencing enzyme efficacy in real-world scenarios.”

We sincerely hope that our revised manuscript could meet the publication standards of **Nature Communications**. Most importantly, we wish to convey to the wider bio-scientific community the significance of assessing enzymatic depolymerization efficiency under conditions pertinent to industry. This perspective, we feel, is pivotal for advancing PETase engineering endeavors for the real-world applicability.

Once again, we appreciate the reviewers’ time and effort given to our manuscript and are very grateful for their invaluable comments that have led to the significant improvement of the presentation of our work.

REVIEWER COMMENTS

Reviewer #1 (Remarks to the Author):

I appreciate the serious effort and additional experiments made to address my criticisms of the original version of the manuscript.

The manuscript is much deeper and more scientific, as the authors elucidated the underlying reasons for the improved performance of TurboPETase. This opens ways for further rational optimization and provides guidelines for the community to apply similar techniques to other enzymes, increasing the broadness and appeal of the work for the community.

In my opinion, the article can be published in the present form.

Reviewer #2 (Remarks to the Author):

I would like to begin by expressing my appreciation for the authors' diligent efforts in conducting additional experiments to address the concerns raised in my previous review. These efforts have notably enhanced the quality of the manuscript. However, there are several aspects that I believe require further attention to ensure the manuscript's coherence and completeness.

Firstly, I observed that some new data have been included only in the rebuttal letter, with references such as Table R1 and Figure R1. These data appear to be of significant value and merit inclusion in the manuscript. I recommend incorporating them partly into the Supplementary Information (partly directly into the main text), with appropriate references and discussions in the main text. This approach would provide a more integrated and comprehensive understanding of the findings.

Regarding the new Figure S14 on PBT degradation and Figure S15 on dual enzyme degradation, I noticed that these figures are not sufficiently discussed (rather not mentioned at all) in the main text. Given the recent interest and publications in high-impact journals on the latter topic regarding S15, it would be beneficial to include a more thorough discussion and relevant citations in the main text if the authors decided to include Fig. S15 in the final manuscript. This would not only enhance the manuscript's relevance but also provide a more complete narrative.

In addition to these points, I have identified a few specific issues that need to be addressed or clarified, possibly with the inclusion of new data, before I can fully endorse this manuscript for publication in Nature Communications:

- 1.) Figures R1 and Table R1 should be included in the Supplementary Information, with references and discussions in the main text. The influence of various buffers at different concentrations on the performance of PETases is a critical aspect of understanding activity differences under laboratory and industrial conditions.
- 2.) Table R3 and Figure R2 should also be included in the Supplementary Information, with brief references and discussions in the main text.
- 3.) The inclusion of Table R2 in the main text is essential, particularly in conjunction with the authors' insightful speculation: "We speculate that due to the much lower Mn of the pretreated PcPET respect to other PET materials, LCCICCG didn't exhibit a rapid decline in degradation rate during the reaction at 72°C when the PET crystallinity reached 20%. However, when the crystallinity increased to 33%, the substantial reduction in the amorphous regions hindered further catalysis." This observation is pivotal as it suggests that once the PET polymer chain length is reduced to a certain threshold (e.g., Mn lower than 9,198, equivalent to a degree of polymerization (DP) range of 45-47), crystallinity is no longer the sole determinant factor for enzymatic degradation. This is a significant finding, as it challenges the previously suggested threshold of less than 20% crystallinity required for enzymatic degradation, which seems to apply primarily to "real" polymers with a DP greater than 100, or Mn more than 20,000. This nuanced understanding aligns well with the viewpoints suggested by Pfaff et al., as well as recent papers by Guo et al. (<https://pubs.acs.org/doi/full/10.1021/acscatal.1c05548>, <https://doi.org/10.1002/cssc.202300742>), which demonstrated that heavily pretreated PET, resulting in short oligomers (DP<20), can be completely crystalline (with only trans conformers) but still remain highly degradable even by IsPETase at 30°C within a few hours. In line with this, I recommend that the authors provide data on the change of crystallinity and molecular weights of the PcPET used in the (high solid loading) degradation experiments shown in Figures 4A and B as time courses. Additionally, including time courses of % crystallinity and Mn&Mw changes under all investigated conditions in parallel to the degradation data in the same figure would greatly contribute to understanding the interplay of varying material properties essential for rapid PET degradation.

4.) The new Figure 2, comparing the degradation performance of various PETases, is a valuable addition. However, I am surprised that the authors have not cited or discussed a recent publication by Carbios (<https://doi.org/10.1021/acscatal.3c02922>), which presents similar experiments under industrial conditions but partly with discrepant outcomes regarding the performance ranking of certain enzymes. Including this reference and discussing the discrepancies between different studies would significantly strengthen the manuscript, particularly in the context of positioning TurboPETase as a promising candidate for industrial applications.

By addressing these points, the authors will significantly enhance the manuscript's contribution to the field of PET degradation, providing a more robust and comprehensive understanding of this important area of research.

Reviewer #3 (Remarks to the Author):

Although the novelty of the study is still rather limited considering the large amount of papers published on PET hydrolyzing enzymes, nevertheless the authors have improved the manuscript and added a lot more data rendering it suitable for publication.

Responses to the reviewers' comments

We sincerely appreciate the careful way the reviewers examined the manuscript. Their valuable suggestions are of great help in refining the manuscript. The manuscript has been revised accordingly. Specific changes are detailed below and are highlighted in yellow in the revised text. We sincerely hope that our revised manuscript could meet the publication standards of Nature Communications.

Point-to-point responses to the reviewer's comments:

Reviewer 2:

Comment 1: I would like to begin by expressing my appreciation for the authors' diligent efforts in conducting additional experiments to address the concerns raised in my previous review. These efforts have notably enhanced the quality of the manuscript. However, there are several aspects that I believe require further attention to ensure the manuscript's coherence and completeness. Firstly, I observed that some new data have been included only in the rebuttal letter, with references such as Table R1 and Figure R1. These data appear to be of significant value and merit inclusion in the manuscript. I recommend incorporating them partly into the Supplementary Information (partly directly into the main text), with appropriate references and discussions in the main text. This approach would provide a more integrated and comprehensive understanding of the findings.

Regarding the new Figure S14 on PBT degradation and Figure S15 on dual enzyme degradation, I noticed that these figures are not sufficiently discussed (rather not mentioned at all) in the main text. Given the recent interest and publications in high-impact journals on the latter topic regarding S15, it would be beneficial to include a more thorough discussion and relevant citations in the main text if the authors decided to include Fig. S15 in the final manuscript. This would not only enhance the manuscript's relevance but also provide a more complete narrative.

Response 1: We extend our sincere gratitude for the valuable and insightful comments throughout the review process. The reviewer's expertise has significantly contributed to enhancing the quality and depth of our work. Following the suggestions, we have integrated the data, previously included in the 1st response letter, into the current revised manuscript. This includes the information in Tables R1-R3 and Figures R1-R2. Regarding the Figure S14 of PBT degradation and Figure S15 of dual enzyme degradation, we have introduced a dedicated section in the revised manuscript, titled "Performance of TurboPETase on alternative substrates and in dual-enzyme systems". Please refer to page 11, lines 347-378, "Previous study has demonstrated limited biodegradation efficiency of *Is*PETase and its variant towards other semiaromatic polyesters, specifically PBT, at 37 °C³⁶. Compared to PET, PBT has slightly lower strength and rigidity, better impact resistance, and a lower glass transition temperature (T_g), which ranges between 37 to 55 °C⁴⁸. In this study, even though 65 °C surpasses the T_g of PBT, thus significantly enhancing the mobility of PBT polymer chains, all of the examined enzymes exhibited substantially reduced degradation efficiency towards PBT films at 65 °C with respect to the degradation of PET (Supplementary Fig. S15). Specifically, TurboPETase yielded higher amounts of hydrolytic products (62.6 μM) than both BhrPETase (27.5 μM) and LCC^{ICCG} (47.1 μM). These results suggested that the active sites of current PET-degrading enzymes were less efficient in binding with the extended aliphatic chains in PBT compared to PET. Consequently, dedicated efforts in enzyme discovery

or tailored engineering are still needed for further improving the depolymerization of new classes of semiaromatic polyesters.

We further explored the application potential of TurboPETase by coupling it with the recently reported BHET hydrolyzing enzyme, BHETase⁴⁹, in a dual-enzyme system. At a low substrate loading (2 g kg⁻¹), the dual-enzyme system effectively doubled the overall yield of the products (sum of BHET, MHET, and TPA), relative to the singular use of TurboPETase (Supplementary Fig. S16). However, an intriguing observation emerged at an elevated PET loading of 30 g kg⁻¹. TurboPETase alone surpassed the yields from most enzyme ratios in the dual-enzyme system, the only exception being the 0.5 mg_{TurboPETase}/g_{PET}:0.1 mg_{BHETase}/g_{PET} ratio. This observed trend echoes a previously report wherein binding modules were added to LCC^{YCCG15}. The fusion enzymes exhibited superior performance at low substrate loadings (< 3 wt% PET). However, as the PET loading intensified (up to 10-20 wt%), they show no sustained advantage over the LCC^{YCCG} domain alone. Nonetheless, in the current investigation, only amorphous PET materials were evaluated. Additional investigations could be conducted to explore the behavior of the dual-enzyme system across PET substrates possessing varied physical morphologies, taking into account factors like crystallinity, accessible surface area, and chemical purity.”

Comment 2: In addition to these points, I have identified a few specific issues that need to be addressed or clarified, possibly with the inclusion of new data, before I can fully endorse this manuscript for publication in Nature Communications:

2.1.) Figures R1 and Table R1 should be included in the Supplementary Information, with references and discussions in the main text. The influence of various buffers at different concentrations on the performance of PETases is a critical aspect of understanding activity differences under laboratory and industrial conditions.

Response 2.1: We sincerely appreciate the reviewer’s suggestion concerning the impact of various buffers at different concentrations on PETase activity. Acknowledging the importance of this aspect, we have added a discussion in the appropriate section of the main text, please refer to pages 5-6, lines 175-182, “In industrial scenarios, enzymatic hydrolysis is typically conducted in water rather than in a dedicated buffered system to simplify downstream processing. Aligned with the industrial preferences, we also evaluated the 12 variants under a low buffer concentration. The results revealed that with the exception M6^{W104G/F243T}, all variants largely retained their degrading activity in 100 mM potassium phosphate buffer (Supplementary Fig. S4 and Supplementary Table S6). Notably, TurboPETase consistently outperformed the other variants under both low and high buffer concentrations, making it the most effective variant for further investigation.”

To maintain the clarity and conciseness of the main text, we have included this data in the Supplementary Materials. The relevant data is now accessible in Supplementary Table S6 and Fig. S4. We hope this amendment aligns with the reviewer’s expectations and are open to any further suggestions or feedback the reviewer may have.

Comment 2.2: Table R3 and Figure R2 should also be included in the Supplementary Information, with brief references and discussions in the main text.

Response 2.2: According to the reviewer’s suggestion, we have relocated Table R3 and Figure R2 to the Supplementary Fig. S18 and Table S11. A brief discussion was also added into the main text. This discussion is seamlessly blended with our analysis of the crystallinity and molecular weight

changes, providing a holistic view of the data and its implications. Please refer to page 12, lines 401-404, "...In the depolymerization of non-melt-quenched PET powders (27.6% crystallinity), we found a significant reduction in degradation performance (Supplementary Fig. S18 and Supplementary Table S11)..."

Comment 2.3: The inclusion of Table R2 in the main text is essential, particularly in conjunction with the authors' insightful speculation: "We speculate that due to the much lower M_n of the pretreated PcPET respect to other PET materials, LCCICCG didn't exhibit a rapid decline in degradation rate during the reaction at 72°C when the PET crystallinity reached 20%. However, when the crystallinity increased to 33%, the substantial reduction in the amorphous regions hindered further catalysis." This observation is pivotal as it suggests that once the PET polymer chain length is reduced to a certain threshold (e.g., M_n lower than 9,198, equivalent to a degree of polymerization (DP) range of 45-47), crystallinity is no longer the sole determinant factor for enzymatic degradation. This is a significant finding, as it challenges the previously suggested threshold of less than 20% crystallinity required for enzymatic degradation, which seems to apply primarily to "real" polymers with a DP greater than 100, or M_n more than 20,000. This nuanced understanding aligns well with the viewpoints suggested by Pfaff et al., as well as recent papers by Guo et al. (<https://pubs.acs.org/doi/full/10.1021/acscatal.1c05548>, <https://doi.org/10.1002/cssc.202300742>), which demonstrated that heavily pretreated PET, resulting in short oligomers (DP<20), can be completely crystalline (with only trans conformers) but still remain highly degradable even by IsPETase at 30°C within a few hours. In line with this, I recommend that the authors provide data on the change of crystallinity and molecular weights of the PcPET used in the (high solid loading) degradation experiments shown in Figures 4A and B as time courses. Additionally, including time courses of % crystallinity and M_n & M_w changes under all investigated conditions in parallel to the degradation data in the same figure would greatly contribute to understanding the interplay of varying material properties essential for rapid PET degradation.

Response 2.3: Thank you for the insightful comment and the opportunity to deepen the discussion in our manuscript. Following your suggestions, we have conducted additional analyses and expanded our discussion, particularly regarding the degradation behavior when PET crystallinity exceeds 20%. This discussion, in line with our initial speculation about the impact of the lower M_n of pretreated PcPET, can be found on page 12, lines 397-417, "Many factors can affect enzymatic attack against the plastics: e.g., crystallinity, chain mobility, molecular size, and surface topography and hydrophobicity¹¹. Among these, the role of low crystallinity in PET degradation has been extensively studied. A crystallinity exceeding 20% has been previously proposed to significantly impede the enzymatic degradation process. In the depolymerization of non-melt-quenched PET powders (27.6% crystallinity), we found a significant reduction in degradation performance (Supplementary Fig. S18 and Supplementary Table S11). However, even when the crystallinity reached 20% after 4 hours of degradation by LCC^{ICCG} at 72 °C, we did not observe a rapid decline in degradation rate. This promoted further investigation through gel permeation chromatography (GPC) analysis, which revealed notably low weight-average (M_w) and number-average (M_n) molecular masses in PcPET powders compared to non-melt-quenched PET powders (Supplementary Table S12). Pfaff et al. showed that LCC^{ICCG} is more efficient in hydrolyzing shorter polymers²⁷. Recent studies also demonstrated that heavily pretreated PET, with a degree of polymerization (DP) less than 20 and high crystallinity, can remain highly degradable^{50, 51}. These findings underscored the importance of factors other than crystallinity in enzymatic degradation.

From the GPC results, we speculate that the low molecular weight of pretreated PcPET may contribute to the maintained degradation rate by LCC^{ICCG} at 72 °C during the first several reaction hours, but a further increase in crystallinity to 33% led to a substantial reduction in the amorphous regions, impeding further enzymatic catalysis.”

Moreover, following your suggestion, we have conducted time-course measurements of the crystallinity and molecular weights of the PcPET used in the high solid loading degradation experiments (Figure 4A). These results, which have been added to new Figure 4, showed only a slight decrease in molecular weights during the reactions. This aligns with previous PET degradation results (*Appl Microbiol Biotechnol* 2009, 84:227–237) and suggested a predominantly surface-level fragmentation of PET. This is different from the degradation pattern of polyvinyl alcohol, which undergo rapid molecular size reduction because the polymer molecules are solved, dispersed, and uniformly susceptible to enzyme attack. For PET, the hydrolysis process involves both endo-type and exo-type hydrolyses concurrently. Kawai et al. suggested that once the surface ester bonds are depolymerized, monomers such as MHET and BHET are produced through exo-type hydrolysis at the ends of fragmented molecules. The sufficiently depolymerized molecules are either released from the PET structure or removed during washing steps preceding GPC analysis, resulting in a marginal change in the overall molecular size of the remaining PET block. Further details of this discussion have been added to the revised manuscript on pages 12-13, lines 417-426, “Moreover, despite more than 90% depolymerization of the PcPET wastes, only a slight decrease in M_w was observed (Fig. 4B), in line with previous PET degradation results hydrolyzed by Cut190³². This suggested a predominantly surface-level fragmentation of PET, involving both endo-type and exo-type processes concurrently. Once the surface ester bonds are depolymerized, monomers such as MHET and BHET are generated from the termini of fragmented polymer chains through exo-type hydrolysis. The sufficiently depolymerized molecules are either released from the PET structure or eliminated during washing steps preceding GPC analysis, leading to marginal changes in the overall molecular size of the remaining PET block³².”

Considering the consistency of the GPC results across all degradation reactions at 65°C for TurboPETase and LCC^{ICCG}, and 72°C for LCC^{ICCG}, and the constraints of the revision timeline, we have only conducted the analyses for Figure 4A with three replicates. We hope this revision aligns with your expectations and addresses your concerns adequately.

Comment 2.4: The new Figure 2, comparing the degradation performance of various PETases, is a valuable addition. However, I am surprised that the authors have not cited or discussed a recent publication by Carbios (<https://doi.org/10.1021/acscatal.3c02922>), which presents similar experiments under industrial conditions but partly with discrepant outcomes regarding the performance ranking of certain enzymes. Including this reference and discussing the discrepancies between different studies would significantly strengthen the manuscript, particularly in the context of positioning TurboPETase as a promising candidate for industrial applications. By addressing these points, the authors will significantly enhance the manuscript's contribution to the field of PET degradation, providing a more robust and comprehensive understanding of this important area of research.

Response: Thanks for highlighting the publication by Carbios, which provides valuable perspectives on enzyme degradation performance under industrially relevant conditions, aligning with the focus of our research. Upon reviewing the article, we observed notable variations in enzyme performance

at different substrate concentrations. Specifically, it showed that HotPETase outperformed PES-H1^{L92F/Q94Y} in specific activity at a lower substrate concentration of 2g L⁻¹, while at higher concentrations, PES-H1^{L92F/Q94Y} demonstrated superior degradation conversion. The authors attributed this performance drop in HotPETase to product inhibition and limited thermostability.

As our experiments in Figure 2, conducted at high solids concentrations (30 g kg⁻¹), yet still below the industrial standard of 200 g kg⁻¹, we acknowledge and respect the data reported in the study by Carbios, as it more closely reflects industrial reaction conditions. However, it is worth noting that even under these industrial conditions, PES-H1^{L92F/Q94Y}'s performance did not surpass that of LCC^{ICCG}. Therefore, we maintain our main conclusion that TurboPETase exhibits superior performance. We have integrated a discussion on this topic into our manuscript, please refer to page 6, lines 206-216, “Recently, similar experiments were conducted to evaluate the degradation performance of FastPETase, HotPETase, PES-H1^{L92F/Q94Y} and LCC^{ICCG} under industrial conditions⁴¹. Their results demonstrated that HotPETase exhibited a higher specific activity compared to PES-H1^{L92F/Q94Y} at a low substrate loading of 2 g L⁻¹. However, as the substrate concentrations increased to 16.5% (w/w) and 20% (w/w), the PET conversion of HotPETase were significantly lower than those of PES-H1^{L92F/Q94Y}. This reduced efficiency at higher substrate loadings is suggested to be attributed to the limited thermostability and other catalytic properties of HotPETase, such as product inhibition. Despite the good performance of PES-H1^{L92F/Q94Y} under industrially relevant conditions, the final depolymerization was still lower than that achieved by LCC^{ICCG}.” and in Figure 2 caption, “*In the investigation conducted by Arnal et al. ⁴¹, the depolymerization efficiency of PES-H1^{L92F/Q94Y} notably surpassed that of HotPETase, yet it remained inferior to LCC^{ICCG}, under industrially relevant substrate loadings of 16.5% (w/w) and 20% (w/w).” This addition will provide readers with a better understanding of the performance of various enzymes under different conditions.

Once again, we wish to extend our deepest appreciation for your invaluable and insightful feedback throughout this review process. Your expertise and thorough guidance have played a pivotal role in refining and enriching our study. Each of your suggestions has been carefully considered and integrated, significantly enhancing the depth, rigor, and alignment of our research with the highest standards of the field. This process has undoubtedly elevated the quality and credibility of our work. We sincerely hope that our revised manuscript could meet the publication standards of Nature Communications.

REVIEWER COMMENTS

Reviewer #2 (Remarks to the Author):

This revised manuscript version demonstrates the potential for significant contribution to the field, particularly with the inclusion of new data and discussions. However, before I can finally endorse its publication in Nature Communications, two critical issues require substantial improvement.

1.) Dual Enzyme System Discussion (Lines 363-378): The authors introduce a dual enzyme system incorporating a BHETase in the main text, suggesting this as an innovative strategy for enhancing degradation performance. However, this concept is not novel, having been previously explored, notably by Barth et al. in 2016 (<https://doi.org/10.1002/biot.201600008>), among other significant studies over the past seven years. These previous studies have utilized different combinations of mesophilic and thermophilic enzymes. To provide a comprehensive and accurate scientific context, the authors must also cite these pivotal works.

<https://doi.org/10.1073/pnas.2006753117>

<https://doi.org/10.1039/D2GC01965E>

<https://doi.org/10.1021/acscatal.2c03772>

<https://doi.org/10.1016/j.checat.2022.11.004>

A quantitative comparison of the efficacy of these previously studied systems with the current study's system is essential. Such a comparison should focus on the overall degradation extent, the synergistic functionality of the dual enzyme systems, and their efficiency in converting inhibitory intermediates like MHET and BHET. This will ensure clarity and prevent any misconceptions among the readers regarding the novelty and efficacy of the proposed system.

2.) Molecular Weight Analysis in Enzymatic PET Depolymerization (Lines 397-426 and Figure 4b): The manuscript presents intriguing data on the development of weight-averaged molecular mass (M_w) during enzymatic PET depolymerization. While this data is insightful for understanding degradation mechanisms, the reliance on M_w as a metric is somewhat limiting. M_w values are disproportionately influenced by larger molecules, and as such, may not accurately reflect changes in molecular sizes. Therefore, I strongly recommend that the authors also include trends of number-average molecular weight (M_n) in Figure 4b (not only in Table S12, where a straightforward comprehension of the time course is not possible). M_n , being the arithmetic mean of molecular mass, correlates with the degree of polymerization (DP) used in NMR analysis by other researchers in this field. This addition would facilitate a more direct and comprehensive comparison, enhancing the manuscript's analytical depth. Additionally, the current discussion on endo- and exo-chain scission mechanisms, particularly within the framework suggested by Kawai et al. (ref. no. 52), appears somewhat elusive and superficial. Expanding this discussion with clearer explanations and deeper insights, coupled with the parallel presentation of M_n development in the main text, would greatly enhance the manuscript's contribution to the understanding of enzymatic PET degradation mechanisms.